# LEARNING TIME-DEPENDENT DENSITY FUNCTIONAL THEORY VIA GEOMETRY AND PHYSICS AWARE LATENT EVOLUTION

## ABSTRACT

We consider using machine learning to simulate time-dependent density functional theory (TDDFT) to predict physical properties of molecules and materials beyond their ground states. In particular, by simulating the electronic response of the system under an external electromagnetic field, the optical absorption spectrum can be calculated using real-time TDDFT (RT-TDDFT), which provides physical information about the excited states and dipole strength function. However, RT-TDDFT simulation requires the direct propagation of electronic wavefunctions of all valence electrons for extended periods, making the process very time-consuming. In this work, we model electron density as volumetric data and train neural networks to map between coarse time steps. To make the model aware of the atomistic environment, we incorporate 3D message passing into the model architecture. Additionally, we use latent evolution to regularize the model towards learning the underlying physics. Our method is termed TDDFTNet. To evaluate our approach, we generate datasets using molecules from the MD17 dataset. Results show that TDDFTNet can learn the time propagation of electron densities accurately and efficiently.

## 1 INTRODUCTION

The ability to understand and predict the behavior of electrons in molecules and materials is crucial for a wide spectrum of natural sciences such as physics, chemistry, materials, and biology. Fundamentally, their interactions are governed by the principles of quantum mechanics, and their quantum states are mathematically described by the many-body wavefunctions from the solutions of the Schrödinger equation. However, their many-body nature poses a significant challenge to achieving the exact many-body wavefunction solution for general molecular or materials system. Various first-principles approaches have been developed in the past to tackle this challenge. For example, according to density functional theory (DFT) (Hohenberg & Kohn, 1964; Kohn & Sham, 1965), electron density itself can completely determine the ground-state properties, making it a very powerful predictive method for studying molecular and materials systems. Runge and Gross rigorously extended the concept of DFT to time-dependent driven systems and established time-dependent density functional theory (TDDFT) (Runge & Gross, 1984) for excited states. TDDFT, especially real-time TDDFT (RT-TDDFT), provides a general framework for investigating linear and nonlinear electron dynamics of molecules and materials with a plethora of applications in physics, chemistry, and biology (Marques & Gross, 2004; Marques et al., 2006), *e.g.*, biological chromophores (Marques et al., 2003), quantum transport (Qian et al., 2006), and plasmonic catalysis (Seemala et al., 2019).

However, first-principles excited-state methods such as RT-TDDFT are highly computationally intensive, with complexity of $O(N^3 \cdot N_t)$ or higher, where $N$ represents the number of electrons in the system and $N_t$ denotes the number of time steps of RT-TDDFT calculation. The steep computational requirement is due to their need to evolve the electron density jointly with the fictitious non-interacting wavefunctions of all $N$ electrons, as well as ensure the stability conditional on sufficiently fine space-time discretizations, limiting the applicability of RT-TDDFT to small systems for a short time scale and creating opportunities for machine learned surrogates to accelerate computation (Zhang et al., 2023). However, despite the immense potential, this direction has been largely under-explored. Here, we introduce *neural TDDFT* to model RT-TDDFT by directly evolving the electron density

on a coarsened grid. Compared to conventional methods for RT-TDDFT built upon mathematical models of quantum mechanics, the neural TDDFT framework achieves efficiency by avoiding explicit evolution of the wavefunctions in addition to learning on a coarser temporal discretization.

We highlight several considerations and proposed solutions for the design of effective neural TDDFT architectures and training frameworks, and employ these insights to propose *TDDFTNet*, the first realization of neural TDDFT. First, neural TDDFT models must learn a representation which integrates both the electron density in the form of volumetric data with the molecule information in the form of a geometric graph. TDDFTNet therefore takes the form of a hybrid convolutional-message passing architecture. Second, in bypassing explicit modeling of the wavefunction, neural TDDFT models are tasked with modeling the evolution of the density with incomplete information. Therefore, generalization of neural TDDFT depends on its ability to implicitly capture the effect of wavefunctions on the density. Thus, we take inspiration from quantum mechanics in designing a physically-inspired regularization imposed on the read-out module for TDDFTNet. Third, we leverage invariances of the Kohn-Sham equations with respect to rotations in order to improve the data efficiency of neural TDDFT. Our experiments utilize TDDFT data generated over 4,000 CPU hours with Octopus (Tancogne-Dejean et al., 2020). To lower the barrier of entry for future works building on our insights, we will release our code and data upon publication.

## 2 REAL-TIME TIME-DEPENDENT DENSITY FUNCTIONAL THEORY

The physical behavior of electrons in atomistic systems is governed by the Schrödinger equation in quantum mechanics, where the many-body electronic wavefunction $\psi\colon \mathbb{R}^{3N} \to \mathbb{C}$ takes the $3N$ coordinate values of all $N$ electrons as input and describes the quantum state of all electrons. For the sake of simplicity, we do not consider the spin degree of freedom here. However, the search space of $\psi$ grows exponentially with $N$. For example, if the 3D space is discretized into $M$ grid points, the number of possible wavefunctions is given by $M^{3N}$.

On the other hand, the electron density offers a more efficient way to describe the electrons. The electron density is directly related to the wavefunctions as

$$\rho(\boldsymbol{r}) = N \int \cdots \int |\psi(\boldsymbol{r}, \boldsymbol{r}_2, \ldots, \boldsymbol{r}_N)|^2 d\boldsymbol{r}_2 \ldots d\boldsymbol{r}_N. \tag{1}$$

In contrast to the $3N$-dimensional wavefunction $\psi$, the electron density $\rho(\boldsymbol{r})$ is a function of 3-dimensional space, regardless of the number of electrons. As a result, the search space for electron density does not increase drastically with more electrons. Fortunately, the Hohenberg-Kohn (HK) theorem proves that electron density is sufficient to describe the ground state of the system, laying the theoretical foundations of DFT (Hohenberg & Kohn, 1964; Kohn & Sham, 1965). The Runge-Gross theorem further extends the HK theorem to the time evolution of electron density, and proves that the physical properties of a many-body system subject to time-varying external potential $v_{\text{ext}}(\boldsymbol{r}, t)$ can be uniquely determined given the time-varying density $\rho(\boldsymbol{r}, t)$, establishing the foundation of TDDFT (Runge & Gross, 1984).

To obtain $\rho(\boldsymbol{r}, t)$, one can map the time-dependent many-body interacting system onto a fictitious time-dependent non-interacting system that yields the same density $\rho(\boldsymbol{r}, t)$ and solve the corresponding time-dependent Kohn-Sham (TDKS) equations, given by

$$i\frac{\partial \phi_j(\boldsymbol{r}, t)}{\partial t} = \left(-\frac{1}{2}\nabla^2 + v_{\text{KS}}[\rho](\boldsymbol{r}, t)\right)\phi_j(\boldsymbol{r}, t), \tag{2}$$

where the Kohn-Sham potential $v_{\text{KS}}[\rho]$ is a functional of $\rho$, consisting of external, Hartree, and exchange-correlation potentials as $v_{\text{KS}}(\boldsymbol{r}, t) \coloneqq v_{\text{ext}}(\boldsymbol{r}, t) + v_{\text{H}}(\boldsymbol{r}, t) + v_{\text{xc}}(\boldsymbol{r}, t)$ (Marques et al., 2006). Furthermore, $\phi_j$ is the time-dependent Kohn-Sham wavefunction for the $j$-th electron. The corresponding electron density is given by

$$\rho(\boldsymbol{r}, t) = \sum_{j=1}^{N} |\phi_j(\boldsymbol{r}, t)|^2. \tag{3}$$

While the mapping to the fictitious system makes the computation more practical, analytical solution of the time-dependent Kohn-Sham equations is intractable, and computational methods and

codes (Marques & Gross, 2004; Marques et al., 2006; Tancogne-Dejean et al., 2020) have been developed for obtaining numerical approximations to $\rho(\boldsymbol{r}, t)$ to solve TDDFT, including linear response TDDFT (Casida, 1995; Jamorski et al., 1996; Andrade et al., 2007) and RT-TDDFT (Castro et al., 2004; Qian et al., 2006).

RT-TDDFT is of particular interest for studying both linear and nonlinear electron dynamics and accessing excited state properties under external fields. However, the stability of numerical schemes depends heavily on sufficiently fine space-time discretizations, resulting in excessive computations. Furthermore, due to the presence of non-interacting electronic wavefunctions in the TDKS equation, the electron density must be evolved jointly with the time-dependent wavefunctions, external potential, and electronic Hamiltonian. Therefore, the computational cost associated with RT-TDDFT increases dramatically with the increasing number of electrons and ions. One potential approach which has thusfar not been explored is the development of machine-learned surrogates for TDDFT. Such a surrogate could improve efficiency by learning to directly evolve the density on a coarser discretization, bypassing explicit simulation of wavefunctions and constraints on grid spacing.

## 3 PROBLEM SETTING

For a given molecule, RT-TDDFT aims to approximate $\rho$ on the computational grid $\mathcal{G} := \mathcal{R} \times \mathcal{T}$, where $\mathcal{R} \subset \mathbb{R}^3$ are a set of $N_x \times N_y \times N_z$ spatial collocation points, and $\mathcal{T} = \{0, t_1, \ldots, t_T\}$ are $T$ equally-spaced temporal collocation points. We denote the numerical approximation to the density at time $t \in \mathcal{T}$ at all spatial collocation points by $\rho_t \in \mathbb{R}^{N_x \times N_y \times N_z}$ which takes the form of a three-dimensional volume, as shown in fig. 1.

Typically, RT-TDDFT methods proceed by first computing the initial non-interacting wavefunctions $\phi_{0,m}$ and density $\rho_0$ by solving the DFT Kohn-Sham equation of the system in ground state at $t = 0$ with $m \in \{1, \ldots, N\}$. Upon the excitation by an external field such as an impulse electric field at $t = 0^+$, the wavefunc-

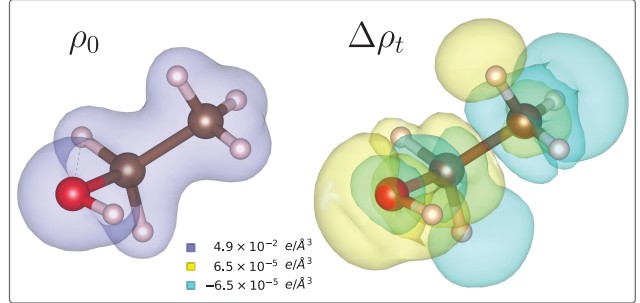

Figure 1: Electron density $\rho_0$ and density difference $\Delta\rho_t := \rho_t - \rho_0$ for ethanol ($C_2H_6O$). Here, $t = 1$ corresponds to the first discrete timestep in the coarsened temporal grid $\tilde{\mathcal{T}}$ following the impulse electric field. Brown, red, and light red spheres represent carbon, oxygen, and hydrogen atoms. The isosurface of $\rho_0$ is indicated by blue. The isosurfaces of $\Delta\rho_t$ are indicated by yellow and cyan for positive and negative isovalues, corresponding to electron accumulation and depletion, respectively.

tions and density are evolved to $\{\phi_{t,m}\}$ and $\rho_t$ by solving the TDKS equations iteratively. Furthermore, to ensure the stability and convergence of the numerical solutions, both the spacing of the grid points in $\mathcal{R}$ and time step size $t_{k+1} - t_k$ must be sufficiently small, limiting the applicability of RT-TDDFT to small systems for a short time scale.

Machine learned surrogate models for TDDFT have been largely unexplored despite their potential to accelerate RT-TDDFT. Here we propose neural TDDFT as efficient alternatives to conventional RT-TDDFT methods by learning to directly evolve $\rho$ on a coarsened grid. Given the initial density $\rho_0$ and corresponding molecule $\mathcal{M}(\boldsymbol{z}, \boldsymbol{C})$, where $\boldsymbol{z} \in \mathbb{Z}^n$ and $\boldsymbol{C} \in \mathbb{R}^{n \times 3}$ are the atom types and coordinates, respectively, we aim to learn a neural TDDFT model $\Phi_\theta$ to compute the evolution of the density as

$$\Phi_\theta\left(\boldsymbol{z}, \boldsymbol{C}, \rho_0\right) = \{\hat{\rho}_t\}_{t \in \tilde{\mathcal{T}} \setminus \{0\}}, \tag{4}$$

where $\tilde{\mathcal{T}} \subset \mathcal{T}$ is a coarsened time discretization of $|\tilde{\mathcal{T}}| = \tilde{T}$ time points such that $\tilde{T} \ll T$, and $\hat{\rho}_t$ denotes the prediction of $\rho_t$. Compared to conventional methods based on mathematical models of quantum mechanics, $\Phi_\theta$ achieves efficiency by avoiding explicit evolution of the wavefunction $\phi$ in addition to the coarser temporal discretization.

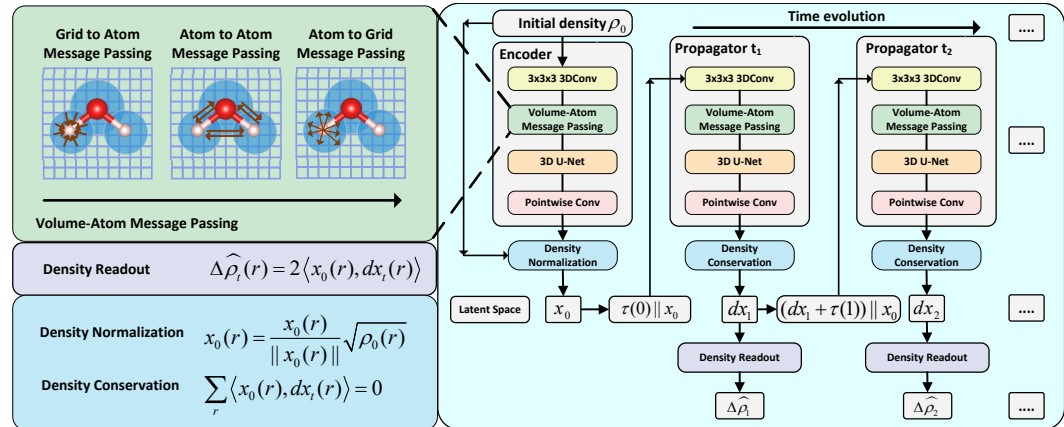

Figure 2: The proposed TDDFTNet framework. The Encoder and Propagator modules in TDDFTNet shown on the right each contain U-Net and message passing layers to integrate the geometric representation of the molecule $\mathcal{M}(\boldsymbol{z}, \boldsymbol{C})$ with the volumetric features $\boldsymbol{x}_t$ corresponding to the electron density $\rho_t$. Additionally, each module includes physics-aware post-processing steps to regularize model predictions towards physical consistency. The Encoder module first lifts the density $\rho_0$ to a latent feature map $\boldsymbol{x}_0$ whose squared norm is constrained such that $\|\boldsymbol{x}_0\|^2 = \rho_0$ to align the latent representation with eq. (3). The Propagator module evolves the density forward from $\rho_t \mapsto \rho_{t+1}$, and is physically constrained for density conservation, as well as to align with eq. (3) via the density readout approximation. The volume-atom message passing submodule is visualized in greater detail on the left and contains three stages of message passing to integrate geometric information with the volumetric feature maps.

## 4 METHOD

### 4.1 LEARNING TIME EVOLUTION OF ELECTRON DENSITIES IN LATENT SPACE

Evolution of quantum systems happens in the space of the wavefunctions, where wavefunctions propagate in time according to the TDKS equations within the TDDFT framework, and physical observables such as the electron density can be computed from the wavefunction at different time steps. Inspired by this physical process, we propose to learn the time evolution of electron densities in latent space. For ease of notation, we use integer subscripts $\rho_t$ to index the discretized time steps on the coarsened temporal grid $\tilde{\mathcal{T}}$ instead of physical time in the remainder of section 4. Given the initial ground state electron density $\rho_0 \in \mathbb{R}^{N_x \times N_y \times N_z}$ corresponding to a molecule $\mathcal{M}(\boldsymbol{z}, \boldsymbol{C})$, we first use an encoder network $f_{\text{enc}}$ to lift it to a feature map $\boldsymbol{x}_0 \in \mathbb{R}^{H \times N_x \times N_y \times N_z}$ with $H$ hidden channels and the same spatial dimensions as

$$\boldsymbol{x}_0 = f_{\text{enc}}\left(\text{emb}(\boldsymbol{z}), \boldsymbol{C}, \rho_0\right). \tag{5}$$

Subsequently, at each time step, the feature map is updated using a propagator network as

$$\boldsymbol{dx}_{t+1} = f_{\text{prop}}\left(\text{emb}(\boldsymbol{z}), \boldsymbol{C}, \boldsymbol{dx}_t + \tau_{\text{emb}}(t), \boldsymbol{x}_0\right), \tag{6}$$

where $\boldsymbol{dx}_{t+1} = \boldsymbol{x}_{t+1} - \boldsymbol{x}_0$ and emb is an atom type embedding. Additionally, we augment the feature map in eq. (6) with a time embedding such that the network is aware of the propagation time. Specifically, $\tau_{\text{emb}}$ encodes $t$ to an $H$-dimensional feature vector via an MLP applied to a sinusoidal positional embedding of $t$ (Vaswani, 2017) which is added to each spatial location in $\boldsymbol{x}_t$. Additionally, to aid the network in retaining information from the initial density, we feed the initial feature map $\boldsymbol{x}_0(\boldsymbol{r})$ to $f_{\text{prop}}$ at each time step. Finally, at each time step, we readout the predicted electron density $\hat{\rho}_t$ with a readout function $f_{\text{readout}}$ as

$$\Delta\hat{\rho}_{t+1} = f_{\text{readout}}(\boldsymbol{x}_0, \boldsymbol{dx}_{t+1}), \tag{7}$$

where $\Delta\hat{\rho}_{t+1} := \hat{\rho}_t - \rho_0$. We use the same network design for both $f_{\text{enc}}$ and $f_{\text{prop}}$, which consists of a message-passing layer (Gilmer et al., 2017) followed by a U-Net (Ronneberger et al., 2015). As we will describe in details below, the message passing layer exchanges the processed information

between the 3D feature map and the atoms, which captures the interactions between the electron cloud and the nuclei. The U-Net processes the feature map with interleaved downsampling steps and convolution layers to gradually extract global information, and then alternatively applies upsampling and convolution layers to refine the local features, with skip-connections between these two stages to preserve high frequency information. We use a U-Net with 3 downsampling and upsampling blocks with an initial channel size of 64. Multi-resolution processing with the U-Net serves to capture multi-scale interactions among electrons. $f_{\text{readout}}$, which maps between the latent feature map and the density, can either be a learnable neural network or a fixed function. In this work, we model $f_{\text{readout}}$ with a physically-motivated fixed function which regularizes feature maps $\boldsymbol{x}_t$ to follow the same semantics as the wavefunction, detailed further in section 4.3. Our pipeline is visualized in fig. 2. In our implementation, we use temporal bundling (Brandstetter et al., 2022b) with 2 time steps, meaning that we predict the latent feature maps for 2 time steps together at each propagation step but use only the second feature map as conditioning for the subsequent propagation.

## 4.2 VOLUME-ATOM GEOMETRIC MESSAGE PASSING

Atom types $\boldsymbol{z}$ and atom coordinates $\boldsymbol{C}$ directly determine the external potential for electrons, and consequently determine the time evolution of the electron density. To incorporate the atomistic environment information, we propose to use geometric message passing between the feature volume and the atoms. Given a 3D point cloud representation of the molecule $\mathcal{M}(\boldsymbol{z}, \boldsymbol{C})$ where each node corresponds to an atom and is assigned a 3D coordinate and feature vector, geometric message passing is employed to compute messages from nodes to update the surrounding volumetric feature vectors based on their relative position, as illustrated by fig. 2. To do so, we jointly consider the grid points $\boldsymbol{r} \in \mathcal{R}$ and atoms comprising the corresponding molecule $\mathcal{M}$ as nodes in a heterogeneous graph. The node features for grid point $\boldsymbol{r}$ is then given by the volumetric feature vector $\boldsymbol{x}_t(\boldsymbol{r})$, while features for the atomistic nodes are initialized dependent on their atom type $\boldsymbol{z}$ and neighboring volumetric features, detailed below. Node features in the heterogeneous graph are then updated based on the received messages from neighboring nodes.

To compute the messages, we employ distance-based geometric message passing as in Schütt et al. (2017). To exchange information between volumetric features and atoms, we use three stages of message passing on the heterogeneous graph. First, we compute the node features $\boldsymbol{v}_i^{(0)}$ for atoms based on their atom types $\boldsymbol{z}$ and gather information from the neighboring region in the volumetric feature map $\boldsymbol{x}_t$ based on the distance between each voxel-atom pair as

$$\boldsymbol{v}_i^{(0)} = \text{emb}(z_i) + \sum_{\boldsymbol{r}_j \in \mathcal{N}_{\text{vol}}(\boldsymbol{C}_i)} \boldsymbol{x}_t(\boldsymbol{r}_j) \odot f^{(0)}(d_{ij}^{(1)}), \tag{8}$$

where $\mathcal{N}_{\text{vol}}(\boldsymbol{C}_i) \subset \mathcal{R}$ denotes the neighboring grid points relative to the coordinates of atom $i$ computed using a distance cutoff, $f^{(0)}$ is a learnable function which embeds the atom-grid distance $d_{ij}^{(1)} := \|\boldsymbol{r}_j - \boldsymbol{C}_i\|$ into an $H$-dimensional feature vector, and $\odot$ denotes channel-wise multiplication between feature vectors. We use a cutoff of 3.5 in our implementation.

Second, we perform $K$ rounds of message passing restricted to the atomistic graph, where the node feature $\boldsymbol{v}_i^{(k)}$ for the $i$-th atom following the $k$-th message passing step for $k > 0$ is given by

$$\boldsymbol{m}_i^{(k)} = \sum_{j \in \mathcal{N}_{\text{atom}}(i)} \boldsymbol{v}_i^{(k-1)} \odot f^{(k)}(d_{ij}^{(2)}) + \boldsymbol{v}_{ij}^{(k-1)}, \quad \boldsymbol{v}_i^{(k)} = \text{MLP}^{(k)}(\boldsymbol{m}_i^{(k)}). \tag{9}$$

Here, $\mathcal{N}_{\text{atom}}(i)$ denotes the indices $j$ of the atom nodes within a distance cutoff of the $i$-th atom and the atom-atom distance $d_{ij}^{(2)}$ is given by $d_{ij}^{(2)} := \|\boldsymbol{C}_i - \boldsymbol{C}_j\|$. We use $K = 3$ in our implementation. A cutoff of 10 is typically used for atom graphs. For our systems, it pratically produces fully-connected graphs. Finally, we distribute the final geometric features $\boldsymbol{v}_j^{(K)}$ back to the volumetric feature map $\boldsymbol{x}_t$ as

$$\boldsymbol{x}_t(\boldsymbol{r}_i) \leftarrow \boldsymbol{x}_t(\boldsymbol{r}_i) + \sum_{\boldsymbol{C}_j \in \mathcal{N}_{\text{vol}}(\boldsymbol{r}_i)} \boldsymbol{v}_j^{(K)} \odot f^{(K+1)}(d_{ij}^{(1)}), \tag{10}$$

where $\mathcal{N}_{\text{vol}}(\boldsymbol{r}_i)$ denotes the atoms within a distance cutoff (=3.5) of the $i$-th gridpoint $\boldsymbol{r}_i$.

### 4.3 PHYSICS-AWARE LATENT REPRESENTATION AND DENSITY READOUT

To regularize the learned latent evolution to align with the underlying physics, we constrain the initial latent feature map and design a map from the latent feature $x_t$ to the predicted density $\hat{\rho}_t$ motivated by quantum mechanics. In particular, inspired by the relation between the wave function $\phi$ and density $\rho$ given in eq. (3), we constrain the square norm of the initial latent feature map at each grid point $r$ to be the same as the initial density as $\|x_0(r)\|^2 = \rho_0(r)$ with density normalization, given by

$$x_0(r) \leftarrow \frac{x_0(r)}{\|x_0(r)\|} \sqrt{\rho_0(r)}. \tag{11}$$

With a similar motivation in mind, we model the density readout at later steps as $\hat{\rho}_t(r) = \|x_t(r)\|^2 = \|x_0(r) + dx_t(r)\|^2$. By doing so, we regularize the latent feature map $x_t$ to have similar semantics as single-electron wavefunctions in DFT, where $x_0$ corresponds to the ground state wavefunction and $dx_t$ corresponds to the wavefunction difference. Importantly, this semantic similarity is kept implicit, as our pipeline does not require supervision of $x_t$ using the ground truth wavefunctions.

As shown in fig. 1, the difference between the target density $\rho_t$ and the initial density $\rho_0$, denoted by $\Delta\rho_t := \rho_t - \rho_0$, is several orders of magnitude smaller than the initial density $\rho_0$. Therefore, instead of directly predicting $\rho_t(r)$, we re-parameterize our prediction target as

$$\Delta\rho_t(r) \approx \hat{\rho}_t(r) - \rho_0(r) = \|x_0(r) + dx_t(r)\|^2 - \|x_0(r)\|^2$$
$$= \|dx_t(r)\|^2 + 2\langle x_0(r), dx_t(r)\rangle. \tag{12}$$

However, due to the small scale of $\Delta\rho_t$, directly using $\|x_0(r) + dx_t(r)\|^2 - \|x_0(r)\|^2$ as the predicted density difference will require $dx_t$ also to be very small compared to $x_0$, in which case adding $dx_t$ and $x_0$ together might lead to unfavorable numerical issues in optimization. We use the following approximation to avoid these problems:

$$\|dx_t(r)\|^2 + 2\langle x_0(r), dx_t(r)\rangle \approx 2\langle x_0(r), dx_t(r)\rangle. \tag{13}$$

In words, we readout the density by using the dot product between $dx_t$ and $x_0$. By doing so, we can conveniently control the scale of $dx_t$ with respect to $x_0$ by scaling the prediction target $\Delta\rho_t$ since now the target is linear in $dx_t$. The approximation in eq. (13) is justified through the assumption that the small magnitude of $\Delta\rho_t$ should correspond to a small magnitude of $dx_t$ due to the semantic similarity of $x_t$ to the wave function. Specifically, we assume that the density $\rho_t$ is continuous in the wave function $\phi_t$ such that a small change in $\rho_t$ implies a small change in $\phi_t$. Additionally, since the total electron density is conserved during time propagation, the total density difference at each time step is zero. To enforce this constraint, we post-process the $dx_t$ as

$$dx_t \leftarrow dx_t - \frac{\sum_r \langle x_0(r), dx_t(r)\rangle}{\sum_r \rho_0(r)} x_0, \tag{14}$$

such that $\sum_r \langle x_0(r), dx_t(r)\rangle = 0$, thereby ensuring that the total predicted electron density does not change compared to $\rho_0$.

### 4.4 LEVERAGING SYMMETRIES OF TDDFT

To improve the data efficiency of neural TDDFT, we leverage symmetries of the Schrödinger equation during training to both augment and canonicalize input-target pairs. For augmentation, we utilize the fact that rotations $R \in SO(3)$ applied to the molecule geometry *about* the direction of the impulse electric field lead to an equal rotation applied to the resulting trajectory. We additionally leverage the equivariance of the resulting trajectories under equal rotations to both the input geometry *and* the polarization direction of external impulse electric field. One of the main quantities of interest in TDDFT simulations is the optical absorption spectrum. In order to compute the spectrum for a general molecule, TDDFT calculations need to be performed multiple times under electric field pulse with various polarizations. Specifically, the direction of the perturbation needs to be varied over the $x$, $y$, and $z$ axes in three independent simulations in order to obtain the total optical absorption spectrum. Therefore, an effective neural TDDFT model should be able to handle all three directions seamlessly. Please refer to appendix E for more details.

## 5 RELATED WORK

**Machine learning density functional theory**: Recently, various deep learning techniques have shown their power in accelerating DFT algorithms. One category of methods take geometric graph networks (Yu et al., 2023; 2024; Gong et al., 2023; Schütt et al., 2019; Li et al., 2022; Unke et al., 2021; Zhong et al., 2023) to predict the final Hamiltonian matrix after the self-consistent loop has converged. These methods require a choice of a set of predefined orbitals in order to obtain the corresponding Hamiltonian matrix, posing the problem of transferability between different orbital sets. Another category of methods leverages deep learning models to predict accurate electron densities (Jørgensen & Bhowmik, 2020; 2022; Cheng & Peng, 2024), thereby accelerating the DFT algorithm. Jørgensen & Bhowmik (2020) learn representations that are independent of the electron density mesh, a favorable property also pursued by more recent works (Fu et al., 2024; Kim & Ahn, 2024; Cheng & Peng, 2024). While these deep learning methods focus on accelerating the DFT algorithm for ground-state systems, their application to the dynamic evolution of electronic states over time remains largely under-explored. Therefore, it is highly desirable to develop machine learning algorithms for accelerating RT-TDDFT.

**Neural PDE solvers**: Solving TDDFT can be naturally framed as solving a partial differential equation (PDE). While machine-learned surrogate models have also been used to accelerate the numerical solution of PDEs (Stachenfeld et al., 2021; Brunton & Kutz, 2023; Kovachki et al., 2023) with applications in fields such as fluid dyanmics (Kochkov et al., 2021; Gupta & Brandstetter, 2023; Lienen et al., 2023) and geophysics (Wu et al., 2022b; Feng et al., 2023; Wen et al., 2023), there has been limited work on surrogates for TDDFT beyond functional learning (Yang & Whitfield, 2023) or direct prediction of energy computed via TDDFT (Pronobis et al., 2018).

While many PDE surrogates rely on convolutional (Lippe et al., 2024; Zhang et al., 2024; Raonic et al., 2024) and message passing architectures (Brandstetter et al., 2022b; Wu et al., 2022a; Toshev et al., 2024), the integration of geometric information within convolutional surrogates has not been explored. Although there has been work on encoding physical information into network architectures (Wang et al., 2020) and loss functions (Li et al., 2021) including for the Schrödinger equation in one spatial dimension (Raissi et al., 2019; Wang & Yan, 2021; Shah et al., 2022; Zeng et al., 2023) and the time-independent DFT Kohn-Sham equation (Nagai et al., 2018; Zepeda-Núñez et al., 2021), the development of deep models of the time-dependent Kohn-Sham equations within the TDDFT framework has been limited. Symmetry priors have also played an important role in the development of neural surrogates for PDEs in architecture design (Wang et al., 2021; 2022; 2023; Bonev et al., 2023; Helwig et al., 2023), loss functions (Akhound-Sadegh et al., 2024), data augmentation (Brandstetter et al., 2022a), and pre-training procedures (Mialon et al., 2023).

## 6 EXPERIMENTS

**Datasets**: We construct our RT-TDDFT datasets using molecular geometries from the MD17 dataset (Chmiela et al., 2017) which consists of thousands of three dimensional structures for several organic molecules. We create datasets for the water, ethanol, and malondialdehyde molecules considered in previous works (Schütt et al., 2019; Yu et al., 2023) by randomly sampling geometries for each molecule from MD17 following a 1,600/200/200 train/valid/test split.

The time-dependent electron density is obtained using a first-principles TDDFT code, Octopus, for each geometry subject to time-dependent electromagnetic fields (Tancogne-Dejean et al., 2020). An impulse field was applied at $t = 0^+$ with an effective kick strength to electronic wavefunctions by 0.01 Å$^{-1}$, where Å denotes Angstroms. We adopt a time-dependent local density approximation of the exchange-correlation energy functional in the Perdew-Zunger form (Perdew & Zunger, 1981) in the RT-TDDFT simulation with Octopus to numerically solve the TDKS equation shown in eq. (2). A uniform three-dimensional real-space mesh is used with a grid spacing of 0.18Å along each Cartesian direction. The corresponding grid cutoff energy is 85.3 Rydberg units. The time integration was performed with a time step of $2.3 \times 10^{-3}\hbar/eV$ for $T =$3,000 total steps from time $t_0 = 0\hbar/eV$ to time $t_T = 6.9\hbar/eV$, where $\hbar$ and eV denote the reduced Planck's constant and electron-Volts, respectively. The electron density was saved at every 50$^{th}$ step for model training such that the neural TDDFT models take a step size of $1.15 \times 10^{-1}\hbar/eV$ for trajectories $\{\rho_t\}_{t \in \tilde{\mathcal{T}}}$, with $\tilde{T} = 60$. We

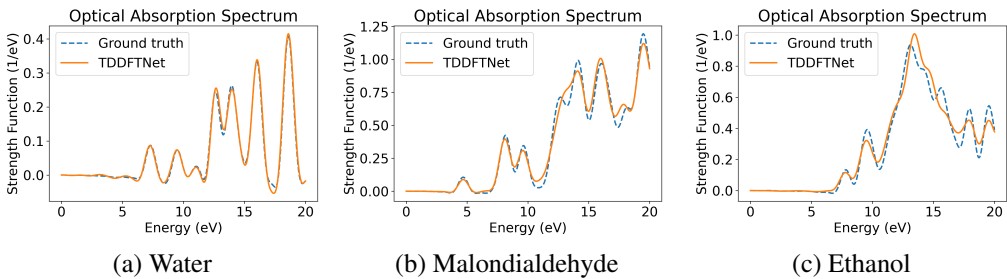

Figure 3: Comparison of optical absorption spectra from the TDDFTNet prediction and the ground-truth calculation.

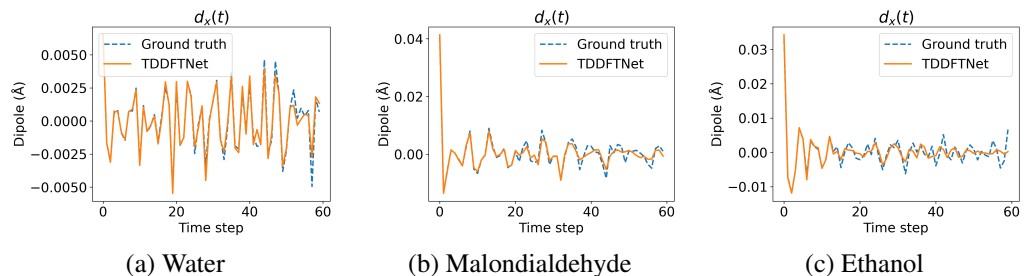

Figure 4: Comparison of predicted dipoles in the $x$ direction with ground-truth values over time. Predicted dipoles are computed from the predicted density as described in section 6, and are used in determining excited energy levels of the molecule, a key quantity of interest. Dipoles in all directions are shown in fig. 7.

center crop the volume around molecule, resulting in a spatial resolution of 48 along all 3 axes for ethanol and malondialdehyde, and 40 for water.

RT-TDDFT simulations were run in parallel on a compute cluster with Intel Xeon 8352Y processors each with 256GB RAM (@2.20GHz), with over 10,000 cores in total. The simulation time highly depends on the complexity of the molecule. For water ($H_2O$), ethanol ($C_2H_6O$), and malondialdehyde ($C_3H_4O_2$), a single trajectory took over 5, 15, and 20 minutes to generate, respectively. For each molecular geometry, three trajectories were generated for the three different electric field polarizations necessary for computing the total optical absorption spectrum, which effectively triples the size of the dataset. Thus, across the 6,000 total simulations required for each molecule, the total CPU hours for generating water, ethanol, and malondialdehyde was greater than 500, 1,500, and 2,000, respectively.

**Multi-stage training**: We use a curriculum training scheme where the number of prediction steps increases during training (List et al., 2024). We divide our training into four stages of increasing difficulty, each with a different number of prediction steps. In the first stage, the model predicts 8 time bundles each consisting of 2 time steps for a total of 16 time steps. In each subsequent stage, the number of bundles is increased by 8, except for in the final stage, where the final number of total predictions is capped at 60, which we find to be sufficient to closely capture the absorption spectrum of the generated trajectories. We train TDDFTNet for 10 epochs for each stage for all three molecules. The cosine annealing learning rate scheduler (Loshchilov & Hutter, 2016) is used within each stage, where the learning rate starts at maximum and gradually decreases to 0 at the end of each stage. At the beginning of each stage the learning rate is restarted at the maximum. We use a maximum learning rate of $2 \times 10^{-4}$ and clip the gradient norm at 1. In addition, we employ gradient checkpointing for the propagator network to reduce GPU memory usage.

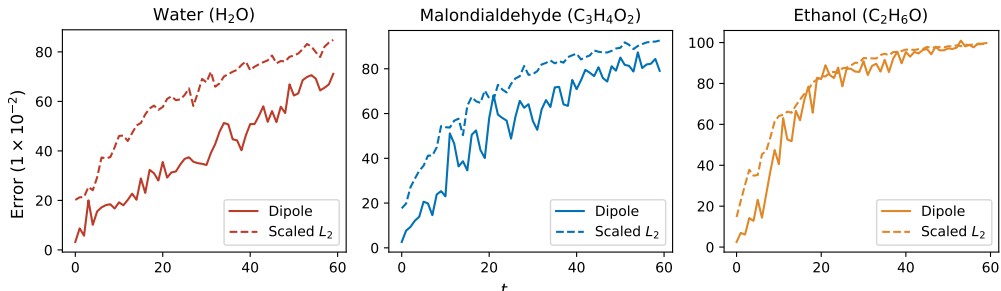

Figure 5: Dipole error and relative error by time step.

**Loss**: Our training objective is a mixture of a Scaled-$L_2$ loss and a Scaled-dipole loss. The Scaled-$L_2$ loss penalizes deviations from the ground truth density difference and is defined as

$$\text{Scaled-}L_2\left(\{\Delta\hat{\rho}_t\}_{t\in[\![1,T]\!]}, \{\Delta\rho_t\}_{t\in[\![1,T]\!]}\right) = \frac{1}{T}\sum_t^T \sqrt{\frac{\sum_{\boldsymbol{r}}\left(\Delta\hat{\rho}_t(\boldsymbol{r}) - \Delta\rho_t(\boldsymbol{r})\right)^2}{\sum_{\boldsymbol{r}}\Delta\rho_t(\boldsymbol{r})^2}}. \quad (15)$$

The Scaled-dipole loss regularizes predictions to have correct dipole moments and is computed as

$$\text{Scaled-dipole}\left(\{\Delta\hat{\rho}_t\}_{t\in[\![1,T]\!]}, \{\Delta\rho_t\}_{t\in[\![1,T]\!]}\right) = \frac{1}{T}\sum_t^T \frac{\|\text{dipole}(\Delta\hat{\rho}_t - \rho_t)\|}{\|\text{dipole}(\rho_t)\|}, \quad (16)$$

where

$$\text{dipole}(\rho) = \sum_{\boldsymbol{r}} \rho(\boldsymbol{r})\boldsymbol{r}\Delta V \in \mathbb{R}^3, \quad (17)$$

and $\Delta V$ represents the real space grid volume. The overall training objective $\mathcal{L}$ is then given by

$$\mathcal{L} = \text{Scaled-}L_2 + \alpha \cdot \text{Scaled-dipole}, \quad (18)$$

where the dipole loss weight $\alpha$ is set to $0.1$.

### 6.1 RESULTS

Results for TDDFTNet on the three molecules are shown in table 1, while the scaled $L_2$ error for density and dipoles at different rollout time steps is shown in fig. 5 and numerically in table 3 in the appendix. In addition to the errors for dipole and density, we also report the overlap between the predicted and the ground truth optical absorption spectra. The absorption spectrum is computed from the Fourier transform of the dipole and is related to the excited energy levels of molecules. Specifically, we compute the spectrum overlap as $\frac{\langle\boldsymbol{\sigma}_{\text{pred}}, \boldsymbol{\sigma}_{\text{gt}}\rangle}{\|\boldsymbol{\sigma}_{\text{pred}}\|\|\boldsymbol{\sigma}_{\text{gt}}\|}$, where $\boldsymbol{\sigma}_{\text{pred}}$ and $\boldsymbol{\sigma}_{\text{gt}}$ denote the predicted and ground truth optical absorption spectrum vectors, where each entry represents a frequency.

Samples of computed optical absorption spectra are shown in fig. 3, while their dipoles in the $x$ direction are shown in fig. 4. As the external electric field is along the $x$ direction, the dipole in the $x$ direction covers the majority of the dipole response. The dipoles in the $y$ and $z$ directions are shown in fig. 7 in the appendix. The absorption spectrum and dipoles of the water molecule closely match the ground truth. Error for malondialdehyde is increased but TDDFTNet is still able to recover most of the significant peaks in the spectrum. The error for the ethanol molecule is larger, potentially because the absorption spectrum is dominated by high frequencies, and the fast varying components in the dipoles may be more difficult for TDDFTNet to capture. Similarly, the dipoles for ethanol are accurate up to a short time window. It is likely that finer temporal resolution could enhance the ability of TDDFTNet to capture the higher frequencies necessary for more accurate modeling of ethanol TDDFT. In appendix D, we analyze the runtime of TDDFTNet compared to a classical TDDFT solver and find a large advantage which would not be substantially comprised by increasing the temporal resolution. Finally, we observe decaying behavior in the predicted dipoles. This may be due to the increased uncertainty from the model at long time predictions. On the other hand, this demonstrates the stability of TDDFTNet over long rollouts, as predictions do not diverge.

**Mixed dataset.** To study the generalization ability to different types of molecules, we create a mixed dataset composed of one third of the trajectories for each of the three molecules. Results in appendix F suggest that TDDFTNet is suitable to be trained across different types of molecules.

Table 1: Average evaluation metrics for water, malondialdehyde, and ethanol molecules.

|  | Dipole Error $(1 \times 10^{-2}) \downarrow$ | Scaled-$L_2$ Error $(1 \times 10^{-2}) \downarrow$ | Spectrum Overlap (%) $\uparrow$ |
|---|---|---|---|
| Water | 39.77 | 61.80 | 99.11 |
| Malondialdehyde | 58.76 | 72.55 | 99.34 |
| Ethanol | 75.21 | 81.07 | 98.78 |

Table 2: Average evaluation metrics for ablation studies on malondialdehyde molecules.

|  | Dipole Error $(1 \times 10^{-2}) \downarrow$ | Scaled-$L_2$ Error $(1 \times 10^{-2}) \downarrow$ | Spectrum Overlap (%) $\uparrow$ |
|---|---|---|---|
| TDDFTNET | **56.79** | **71.45** | **99.34** |
| NO-MP | 67.58 | 78.04 | 98.86 |
| MP-U-NET | 59.31 | 74.18 | 99.25 |
| MP-U-NET-ONE-SHOT | 62.41 | 77.51 | 99.15 |
| MP-U-NET-AR | 137.68 | 86.92 | 95.74 |

## 6.2 ABLATION STUDIES

We next perform an ablation study to evaluate the effectiveness of various components of TDDFTNet:

- NO-MP: removes message passing blocks.
- MP-U-NET: removes the physics-aware components, including the density normalization on encoder's output $x_0$, as well as the physics-based density readout. The density differences $\Delta \rho_t$ are read out using a linear layer. The mean of the output is subtracted from each time step to ensure the total density difference is zero at each time step.
- MP-U-NET-ONE-SHOT: same as MP-U-NET, additionally replaces latent evolution with one-shot prediction, where a linear layer whose output channel is equal to the number of prediction time steps is used to yield the output of all time steps using a single forward pass of the model.
- MP-U-NET-AR: same as MP-U-NET, additionally replaces latent evolution with one-step-ahead prediction. During training, the model learns to predict the density difference at the next time step $\Delta \rho_{t+1}$ from the density difference at current time step $\Delta \rho_t$ together with the initial density $\rho_0$. During evaluation, rollouts are produced autoregressively.

Numerical ablation study results on Malondialdehyde are presented in table 4. Results by time step are presented in fig. 6 and numerically in table 4 in the appendix. The observed detriment of removing message passing demonstrates the importance of capturing interactions between the electron cloud and the nuclei. Furthermore, the integration of physics-aware components improves metrics across the board, while latent evolution with a training curriculum proves to be a more effective training strategy than one-shot or one-step prediction. We provide more details about autoregressive models in appendix G and the effect of using U-Net in appendix H.

## 7 CONCLUSION

We study neural TDDFT and develop a method, known as TDDFTNet, that simulates and accelerates TDDFT using machine learning. The architecture of TDDFTNet integrates geometric information from the three-dimensional molecular graph in modeling the volumetric electron density by leveraging a unified, heterogeneous graph representation of these entities. Physics-aware training of TDDFTNet regularizes the model to maintain a greater degree of physical consistency. Results show that TDDFTNet achieves promising performance with dramatic acceleration. Our framework can be extended to crystalline materials and amorphous systems, with great potential for studying excited state dynamics.

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

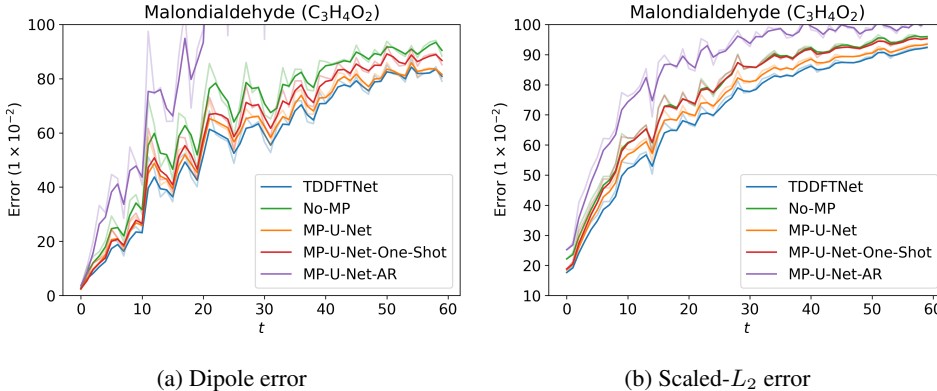

(a) Dipole error
(b) Scaled-$L_2$ error

Figure 6: Ablation studies for TDDFTNet. Original curves (with shading) are smoothed using exponential moving average for better visibility.

## A  OPEN DIRECTIONS

The introduction of deep learning into the realm of TDDFT has created many potential lines of work, several of which we discuss here. Volumetric modeling in 3 spatial dimensions invites innovation towards scalable solutions for more complex systems, including macromolecules, periodic structures, and larger molecules, although generation of a machine learning-scale dataset for such systems will require substantial computational resources. However, the existence of a large-scale mixed molecule dataset would enable development of neural TDDFT frameworks that generalize over molecule type, which we explore further in appendix F. Various applications of TDDFT may call for time-varying external potentials or varying excitation types, in which case the development of representations capturing the effect of the potential will be vital for faithful modeling. The extension of continuous models of the electron density, as explored for static DFT by Jørgensen & Bhowmik (2020); Fu et al. (2024); Cheng & Peng (2024); Kim & Ahn (2024), is also a promising direction.

## B  ETHICS AND REPRODUCIBILITY STATEMENT

Neural TDDFT has the potential to accelerate new discoveries in biology and material science. To lower the barrier of entry for future works building on our insights, we will release our code and data upon publication.

## C  RESULTS

We present numerical results for our main experiments in table 3, numerical results for ablation study in table 4, and plots of predicted dipoles and their ground-truth on all $x$, $y$ and $z$ axes in fig. 7. Note that the scale for the dipole values along the $y$-axis and $z$-axis are relatively smaller compared to those along the $x$-axis.

Table 3: Dipole error and relative error by time step.

| Time Step | Dipole Error ($1 \times 10^{-2}$) | | | | | | | Scaled $L_2$ Error ($1 \times 10^{-2}$) | | | | | | |
|---|---|---|---|---|---|---|---|---|---|---|---|---|---|---|
| | 1 | 10 | 20 | 30 | 40 | 50 | 60 | 1 | 10 | 20 | 30 | 40 | 50 | 60 |
| Water | 3.21 | 16.72 | 28.02 | 34.78 | 46.26 | 66.90 | 71.13 | 20.21 | 41.46 | 56.57 | 69.09 | 76.10 | 78.02 | 84.84 |
| Malondialdehyde | 2.61 | 25.34 | 40.13 | 64.17 | 75.02 | 79.13 | 79.07 | 17.64 | 54.40 | 70.30 | 77.26 | 86.05 | 89.08 | 92.61 |
| Ethanol | 2.48 | 47.47 | 82.73 | 85.63 | 89.86 | 96.25 | 99.78 | 14.70 | 61.43 | 81.91 | 90.16 | 95.89 | 97.67 | 99.24 |

Table 4: Ablation study dipole error and relative error by time step.

| Time Step | Dipole Error ($1 \times 10^{-2}$) | | | | | | | Scaled $L_2$ Error ($1 \times 10^{-2}$) | | | | | | |
|---|---|---|---|---|---|---|---|---|---|---|---|---|---|---|
| | 1 | 10 | 20 | 30 | 40 | 50 | 60 | 1 | 10 | 20 | 30 | 40 | 50 | 60 |
| TDDFTNet | **2.61** | **25.34** | **40.13** | **64.17** | **75.02** | **79.13** | **79.07** | **17.64** | **54.40** | **70.30** | **77.26** | **86.05** | **89.08** | **92.61** |
| No-MP | 2.82 | 37.24 | 46.76 | 76.99 | 86.69 | 88.43 | 88.37 | 22.16 | 63.74 | 77.16 | 85.17 | 92.72 | 93.94 | 96.03 |
| MP-U-Net | **2.35** | 29.05 | 43.09 | 66.25 | 76.23 | 80.34 | 80.23 | 18.53 | 59.70 | 73.08 | 80.56 | 88.42 | 90.68 | 93.71 |
| MP-U-Net-One-Shot | 2.46 | 30.40 | 42.84 | 70.64 | 81.15 | 84.88 | 85.20 | 18.82 | 62.71 | 77.61 | 85.19 | 92.24 | 93.14 | 95.47 |
| MP-UNet-AR | 3.62 | 49.00 | 90.89 | 227.35 | 232.54 | 237.05 | 227.95 | 25.21 | 77.38 | 91.94 | 94.50 | 103.48 | 98.91 | 101.91 |

## D  TIME COMPARISON

In table 5, we compare the runtime of TDDFTNet to the classical TDDFT method implemented in Octopus on the ethanol dataset. Note that TDDFTNet was run with a batch size of 4 on a single A100 GPU. We find TDDFTNet offers a large speed up relative to the classical solver.

Table 5: Time comparison on the Ethanol dataset.

| Method | Time |
|---|---|
| Octopus | >15 min. |
| TDDFTNet | 1.78 sec. |

## E  ROTATION EQUIVARIANCE AND DATA AUGMENTATION

To improve the data efficiency of neural TDDFT, we leverage symmetries of the Schrödinger equation during training to both augment and canonicalize input-target pairs. For augmentation, we utilize the fact that rotations $R \in \mathrm{SO}(3)$ applied to the molecule geometry *about* the direction of the impulse electric field lead to an equal rotation applied to the resulting trajectory. Thus, for the molecule-density pair $\left(\{\rho_t(\boldsymbol{r})\}_{t\in\hat{\mathcal{T}}}, \mathcal{M}(\boldsymbol{z}, \boldsymbol{C})\right)$, the rotated pair $\left(\{\rho_t(\boldsymbol{R}^{-1}\boldsymbol{r})\}_{t\in\hat{\mathcal{T}}}, \mathcal{M}(\boldsymbol{z}, \boldsymbol{R}\boldsymbol{C})\right)$ also corresponds to a solution of the Kohn-Sham equations. Therefore, during training, we apply random rotation sampled from $\{\boldsymbol{I}, \boldsymbol{R}^{90^{\circ}}, \boldsymbol{R}^{180^{\circ}}, \boldsymbol{R}^{270^{\circ}}\}$ about the direction of the impulse electric field to inputs and targets, effectively expanding the size of the training data by a factor of four. Rotations by arbitrary angles can also be employed, with interpolation used to resolve inconsistencies with the rotated density and the computational grid.

We additionally leverage the equivariance of the resulting trajectories under equal rotations to both the input geometry *and* the polarization direction of external impulse electric field. One of the main quantities of interest in TDDFT simulations is the optical absorption spectrum. In order to compute the spectrum for a general molecule, TDDFT calculations need to be performed multiple times under electric field pulse with various polarizations. Specifically, the direction of the perturbation needs to be varied over the $x$, $y$, and $z$ axes in three independent simulations in order to obtain the total optical absorption spectrum. Therefore, an effective neural TDDFT model should be able to handle all three directions seamlessly. One possibility could be to condition the model on the direction, however, this would cost valuable model capacity. Instead, we optimally leverage the mutual information between the data from the three polarizations by canonicalizing inputs and targets such that the direction of the perturbation is invariant. Specifically, given a set of three density trajectories $\{\rho_t^x(\boldsymbol{r}), \rho_t^y(\boldsymbol{r}), \rho_t^z(\boldsymbol{r})\}_{t\in\hat{\mathcal{T}}}$ corresponding to the molecule $\mathcal{M}(\boldsymbol{z}, \boldsymbol{C})$ excited by impulse electric field from the $x$, $y$, and $z$ direction, respectively, we apply the deterministic rotations $\boldsymbol{R}_y$ and $\boldsymbol{R}_z$ to the molecule-density pairs as

$$\left(\{\rho_t^y(\boldsymbol{R}_y^{-1}\boldsymbol{r})\}_{t\in\hat{\mathcal{T}}}, \mathcal{M}(\boldsymbol{z}, \boldsymbol{R}_y\boldsymbol{C})\right) \qquad \left(\{\rho_t^z(\boldsymbol{R}_z^{-1}\boldsymbol{r})\}_{t\in\hat{\mathcal{T}}}, \mathcal{M}(\boldsymbol{z}, \boldsymbol{R}_z\boldsymbol{C})\right) \tag{19}$$

such that the polarization of the impulse electric field is the same as for the $x$-direction given by $(\rho_t^x(\boldsymbol{r}), \mathcal{M}(\boldsymbol{z}, \boldsymbol{C}))$. To compute the optical absorption spectrum, the inverse rotations are then applied to the predicted trajectories as

$$\{\hat{\rho}_t^y(\boldsymbol{R}_y\boldsymbol{r})\}_{t\in\hat{\mathcal{T}}\setminus\{0\}} \qquad \{\hat{\rho}_t^z(\boldsymbol{R}_z\boldsymbol{r})\}_{t\in\hat{\mathcal{T}}\setminus\{0\}} \tag{20}$$

to return them to their original orientation.

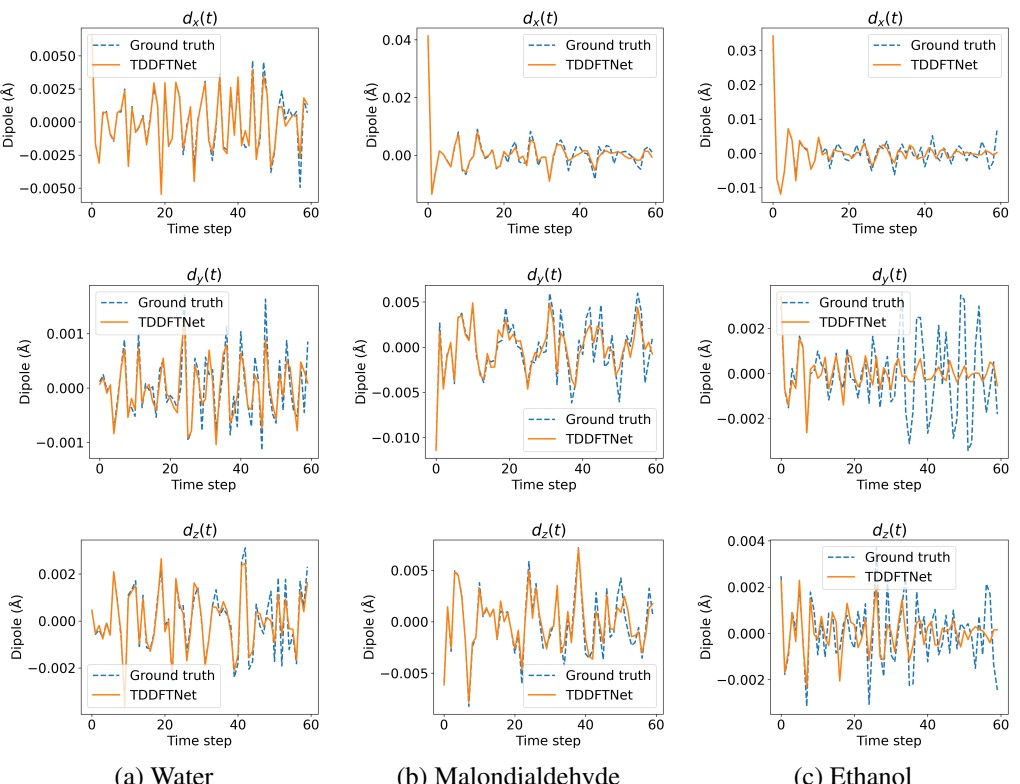

(a) Water     (b) Malondialdehyde     (c) Ethanol

Figure 7: Comparison of predicted dipoles with ground-truth values over time along the $x$-axis, $y$-axis, and $z$-axis.

## F  MIXED DATASET

The training set is composed of 534 molecules taken from each of the three datasets. The validation and test sets are composed of 67 molecules taken from each of the three datasets. For each molecule, we use data from all 3 electric field directions. We use a spatial resolution of 48 for all 3 molecules.

Samples of optical absorption spectrum are shown in fig. 8, and average metrics are shown in table 6. The metrics for different molecules are averaged on their respective subsets, while the overall results are evaluated on all three types of molecules and represents the average performance.

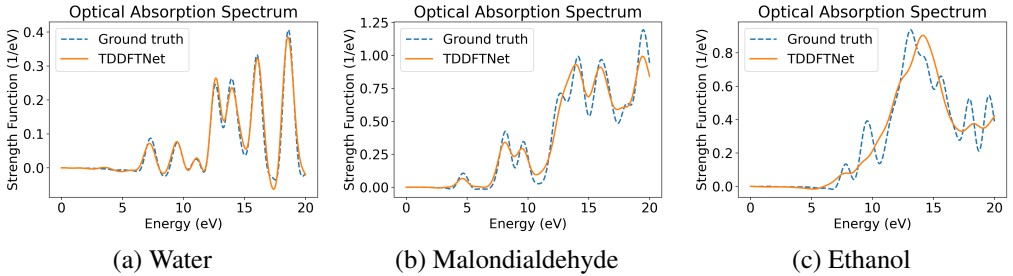

(a) Water     (b) Malondialdehyde     (c) Ethanol

Figure 8: Comparison of optical absorption spectra from the TDDFTNet prediction and the ground-truth calculation using the model trained on the mixed dataset.

Table 6: Average evaluation metrics of TDDFTNet on the mixed dataset.

| | Dipole Error $(1 \times 10^{-2})\downarrow$ | Scaled-$L_2$ Error $(1 \times 10^{-2})\downarrow$ | Spectrum Overlap (%) $\uparrow$ |
|---|---|---|---|
| Water | 53.32 | 65.28 | 97.83 |
| Malondialdehyde | 69.55 | 80.66 | 98.69 |
| Ethanol | 84.55 | 88.79 | 97.66 |
| Overall | 69.14 | 78.24 | 98.06 |

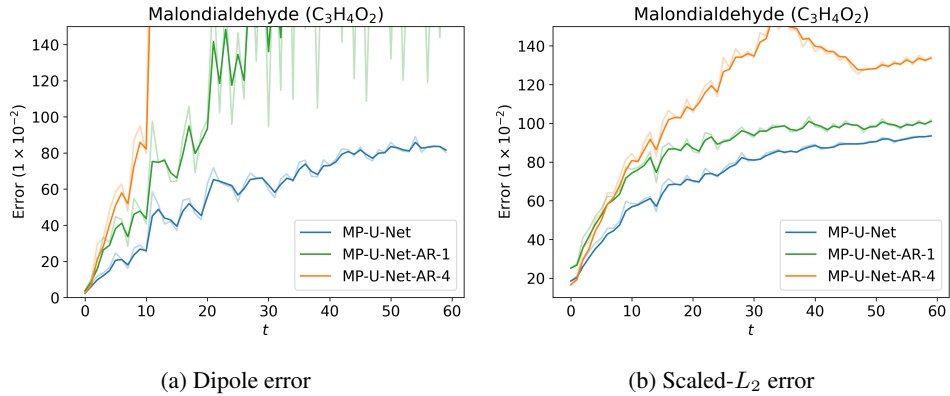

(a) Dipole error

(b) Scaled-$L_2$ error

Figure 9: Ablation studies for autoregressive models. Original curves (with shading) are smoothed using exponential moving average for better visibility.

## G AUTOREGRESSIVE MODEL

In autoregressive prediction, the model is trained to predict the next steps based on the immediate previous steps. We additionally condition the prediction on the initial density since it contains important background information. Formally, we condition on $[\rho_0, \Delta\rho_{t-h+1}, \ldots, \Delta\rho_t]$ to predict $[\Delta\rho_{t+1}, \ldots, \Delta\rho_{t+f}]$, where $\rho_0$ is the initial density, $h$ is the number of historical time steps, and $f$ is the number future time steps. We chose $f = 2$ and tested with $h = 4$ and $h = 1$.

The results are shown in fig. 9. A sample of optical absorption spectrum for the Malondialdehyde molecule using $h = 1$ is shown in fig. 10. The results show that autoregressive prediction yields significantly larger rollout errors compared to latent evolution or one-shot prediction (see fig. 6 in main text). Besides error accumulation during rollout, one of the other reasons that autoregressive prediction fails may be because the Markov assumption does not hold, as the electron density at one time step cannot fully determine the quantum state.

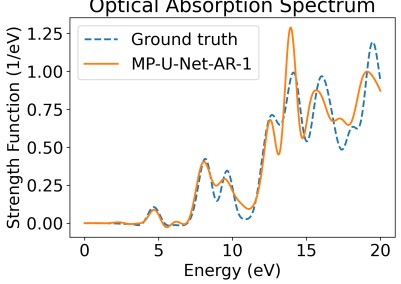

Figure 10: Autoregressive model with $h = 1$ on Malondialdehyde

# H   ABLATION ON U-NET

To study the effectiveness of multi-scale processing in the U-Net, we created an ablated U-Net with only one downsampling / upsampling step with the number channels multiplied by 2 after downsampling. More concretely, for the downsampling branch, we first apply a convolution block at the highest resolution, then apply the downsampling, and apply another convolution block afterward. The upsampling branch is similar. We also created a ResNet with exactly the same architecture as the ablated U-Net by removing all downsampling / upsampling steps. The resulting U-Net and ResNet have the same number of parameters (12M). We then train them on the mixed dataset using one-shot prediction without message passing blocks.

The results are shown in table 7. Although ResNet results in slightly better Scaled-$L_2$ error, U-Net gives better dipole error. This suggests that multi-resolution processing is helpful in capturing global context necessary for accurate modeling of dipoles. We also notice that ResNet uses 1.5x training time and 1.3x GPU memory compared to U-Net. The computational and memory efficiency of U-Net is important for processing 3D volumetric data. Overall, the ability to learn global properties and the computational efficiency make U-Net suitable for learning TDDFT.

Table 7: Average evaluation metrics for U-Net and ResNet on mixed dataset.

| | Dipole Error ($1 \times 10^{-2}$)↓ | Scaled-$L_2$ Error ($1 \times 10^{-2}$)↓ | Spectrum Overlap (%)↑ |
|---|---|---|---|
| U-NET-ONE-SHOT | **79.07** | 83.77 | **95.74** |
| RESNET-ONE-SHOT | 80.05 | **83.38** | 95.36 |

# I   DATASET AND PREDICTION VISUALIZATION

We plot the input electron density of a sample Malondialdehyde molecules in fig. 11, and the electron density rollout in fig. 12. The initial electron densitiy and the rollout of density differences and are first cropped to the center 12 voxels along $y-$axis, and then summed along $y-$axis to produce a 2D plot. The absolute errors are first computed for the entire volume and then cropped and summed in the same way.

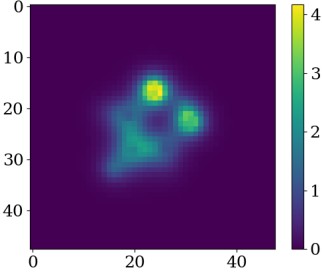

Figure 11: Initial electron density of malondialdehyde molecule.

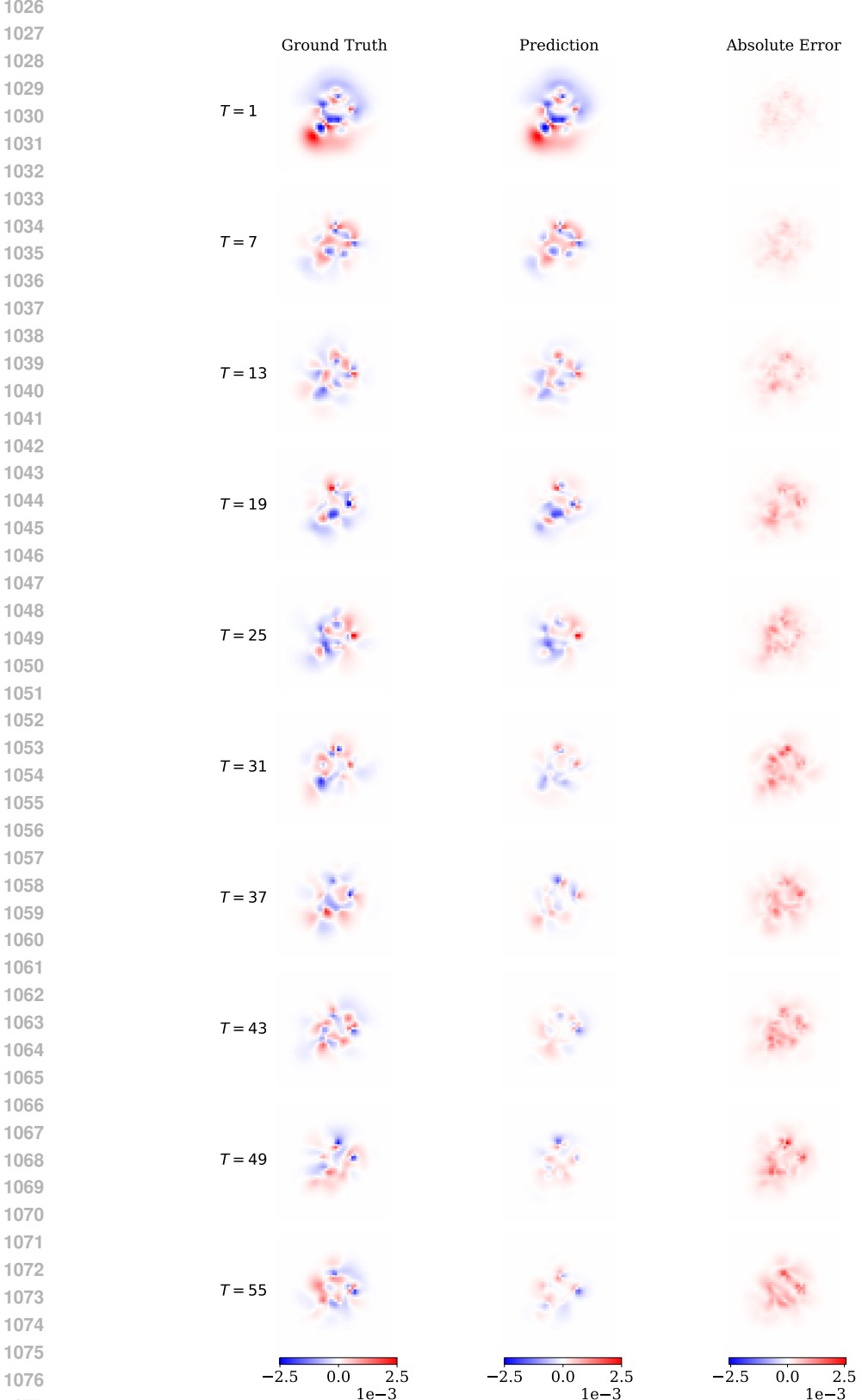

Figure 12: Rollout of electron density differences of malondialdehyde molecule.

