# OpenReview forum: "Learning Time-Dependent Density Functional Theory via Geometry and Physics Aware Latent Evolution"
_ICLR.cc/2025/Conference — Submitted to ICLR 2025_

### Official Review · Reviewer_t8k9 · 2024-10-29

**Soundness:** 2
**Presentation:** 2
**Contribution:** 2
**Rating:** 5
**Confidence:** 3

**Summary:**

This paper proposes an algorithm to simulate the time-dependent density functional theory (TDDFT) and predict the properties of molecules. Compared to traditional simulators, NN-based methods can speed up the process. By using encoder, the method encoders atom types and atom coordinates into latent space, then with propagators to predict the density function as time evolution goes.

**Strengths:**

This paper does an interdisciplinary research for simulating the TDDFT with a neural network based method and provides some analysis about how to fit the network design with respect to specific physical science constraints.

**Weaknesses:**

1. This paper just simply combines neural networks into the physical sciences problems for predicting TDDFT for molecules. Due to the lack of comparison with other learning based methods and insufficient experiment results, I don’t see the novelty and effectiveness of this method from the learning perspective. Maybe this work is more appropriate for some physical science journals.
2. This paper only does experiments on a very limited number of molecules and only provides in-distribution testing for these samples. I think the value of this method would be limited if it needs to train for each molecule individually.
3. There is no comparison for this method with other state-of-art work but I think using neural networks to predict for molecules is a very popular topic.

**Questions:**

1. Why do the experiments only test on three molecules? Does each molecule need an independent model? And what is the result if the model is trained on one type of molecule but tested on another unseen molecule?
2. What about the comparison of TDDFTNet with other state-of-art learning based methods to predict properties of the molecules?
3. Please clarify what the three subplots in figure 4 is about.

---

> ### Author Response · Authors · 2024-11-24
>
> ## W1, W3, and Q2
>
> > W1. This paper just simply combines neural networks into the physical sciences problems for predicting TDDFT for molecules. Due to the lack of comparison with other learning based methods and insufficient experiment results, I don’t see the novelty and effectiveness of this method from the learning perspective. Maybe this work is more appropriate for some physical science journals.
>
> > W3. There is no comparison for this method with other state-of-art work but I think using neural networks to predict for molecules is a very popular topic.
>
> > Q2. What about the comparison of TDDFTNet with other state-of-art learning based methods to predict properties of the molecules?
>
>
> Our proposed neural TDDFT framework assumes that to obtain time-dependent quantum properties, such as the dipole moments $\operatorname{dipole}(\rho)$, we must temporally evolve the density and then compute these properties from the predicted density. This is more complicated compared to well-established neural methods for molecular property prediction that learn a direct mapping from the molecule to the property of interest. It is therefore of vital importance to explore this direct mapping approach to justify our proposed framework, and we thank you for raising this point. We summarize the experimental setup and results below and we will include the results in the revised manuscript.
>
> Given a Malondialdehyde molecule $\mathcal{M}(\boldsymbol{z},\mathbf{C})$, where $\boldsymbol{z}\in\mathbb{Z}^n$ and $\mathbf{C}\in\mathbb{R}^{n\times 3}$ are the atom types and coordinates, respectively, we aim to train a GNN model $\Phi_\theta$ that can directly predict $\{\operatorname{dipole}(\rho_t)\}_{t=1}^{\tilde{T}}\in\mathbb{R}^{3\times\tilde{T}}$  corresponding to the molecule, where a formal definition of the dipole moments can be found shortly before Section 6.1 in our manuscript. As the GNN backbone, we consider a variety of molecular property prediction models.
>
> - **SchNet** (Schütt et al., 2018): learns molecular properties using atom types and node features and inter-atomic distance as features. We essentially employ the same design to do atom-atom message passing in TDDFTNet.
> - **SphereNet** (Liu et al., 2021): additionally considers angles from any three neighboring atom, as well as torsion angles from any four neighboring angles. Although it is not a large model by current standards, it usually offers state-of-the-art performance on small-scale datasets such as the dataset in our paper.
>
> One major issue of directly applying these property prediction models in our setting is that these models usually assume rotation-invariance. However, in our problem, the molecule interacts with a electric field pulse which comes from one pre-defined direction. As a result, the resulting dipole values are no longer invariant when a molecule undergoes a 3D rotation.
>
> Therefore, to prevent these models from enforcing incorrect symmetries, we include the absolute position of atoms as an additional input node features so that the models are no longer $\operatorname{SO}(3)$-invariant. As our neural TDDFT models are additionally conditioned on the ground state density $\rho_0\in \mathbb{R}^{N_x\times N_y \times N_z}$, for SphereNet and SchNet, we experiment with and without conditioning GNN models on an embedding $\boldsymbol{x}_0\in\mathbb{R}^H$ of $\rho_0$, where the embedding is extracted using the down-sampling part of a U-Net followed by a global pooling operation.

---

> ### Author Response · Authors · 2024-11-24
>
> The results are summarized in the below table, where SchNet and SphereNet denote the original GNNs with rotation-invariance. SchNet-coord and SphereNet-coord are the GNN models with additional coordinate features to break the rotation-invariance. Finally, SchNet-coord-cnn and SphereNet-coord-cnn denote the models additionally conditioning on the initial density features $\boldsymbol{x}_0$ extracted using CNNs.
>
> As can be seen, the original GNN models enforcing rotation-invariance perform poorly on the task, with dipole errors over 90%. In comparison, the -coord models perform significantly better, where SphereNet can produce dipole error close to TDDFTNet ($60.31\times10^{-2}$).
>
> To further push the performance of the GNN models, we increase the the hidden dimension and the number of layers of SphereNet. In SphereNet-w-coord, we increase the hidden dimension from 128 to 256. In SphereNet-d-coord, we increase the number of GNN layers from 4 to 6. In SphereNet-wd-coord, we incorporate both modifications by using a hidden dimension of 256 and 6 GNN layers. Each variant outperforms the original SphereNet. Notably, SphereNet-wd-coord yields slightly better performance than TDDFTNet.
>
> Finally, conditioning on the initial density improves the performance slightly for SphereNet-w-coord and SphereNet-d-coord but not for SphereNet-wd-coord. On the other hand, TDDFTNet can effectively learn information from the initial density with a simple GNN such as SchNet, highlighting the effectiveness of our proposed volume-atom message passing mechanism, which is one of the major novelties of our method from the learning perspective, the benefits of which are demonstrated by our ablation study in Figure 6.
>
>
> | Model                   | Dipole Error ($1\times10^{-2}$)$\downarrow$ | Spectrum Overlap (\%)$\uparrow$ |
> |-------------------------|---------------------------------|-----------------------|
> | TDDFTNet                |   58.76                         |   99.34               |
> | SchNet                  |   90.56                         |   96.53               |
> | SchNet-coord            |   68.61                         |   98.87               |
> | SphereNet               |   90.22                         |   96.62               |
> | SphereNet-coord         |   60.31                         |   99.31               |
> | SphereNet-w-coord       |   59.84                         |   99.34               |
> | SphereNet-d-coord       |   59.26                         |   99.34               |
> | SphereNet-wd-coord      |   58.26                         |   99.39               |
> | SphereNet-w-coord-cnn   |   59.30                         |   99.36               |
> | SphereNet-d-coord-cnn   |   59.17                         |   99.37               |
> | SphereNet-wd-coord-cnn  |   58.56                         |   99.39               |
>
>
> Besides SchNet and SphereNet, we also tried a large GNN foundation model -- **JMP** (Shoghi et al., 2023), which is pretrained on a diverse set of molecular property predictions datasets. However, despite our best efforts, it does not outperform SphereNet and results in a dipole error of 68.07% even when conditioned on atom coordinates. Nevertheless, we do believe a pretrained foundation model could potentially yield an even better result given more tuning.
>
> Lastly, we would like to revisit the point that compared to direct prediction of dipoles, electron density is more a general target within the TDDFT framework. By modeling the time evolution of electron density, we are able to derive more properties, including higher-order multipoles, defined as $\int \rho(r) r^d dr$, with $d$ denoting the order, without re-creating the dataset or re-training the model.
>
>
> Liu, Yi, et al. "Spherical message passing for 3d graph networks." arXiv preprint arXiv:2102.05013 (2021).
>
> Schütt, Kristof T., et al. "Schnet–a deep learning architecture for molecules and materials." The Journal of Chemical Physics 148.24 (2018).
>
> Shoghi, Nima, et al. "From molecules to materials: Pre-training large generalizable models for atomic property prediction." arXiv preprint arXiv:2310.16802 (2023).

---

> ### Author Response · Authors · 2024-11-24
>
> ## W2 and Q1
>
> > W2. This paper only does experiments on a very limited number of molecules and only provides in-distribution testing for these samples. I think the value of this method would be limited if it needs to train for each molecule individually.
>
> > Q1. Why do the experiments only test on three molecules? Does each molecule need an independent model? And what is the result if the model is trained on one type of molecule but tested on another unseen molecule?
>
>
> There are many applications in physics, chemistry, and biology in which TDDFT must be solved many times for different geometries of the same molecule, such as biological chromophores, quantum transport, and plasmonic catalysis. Small differences in the geometry can lead to distinct time evolutions of the density, and thus, the number of geometries to be explored in determining viability can easily extend beyond thousands and even orders of magnitude above. In these cases, neural surrogate models can save thousands of hours of computation time, as shown in the "Time Comparison" appendix, and therefore hold great value even if only trained for a single molecule.
>
>
> Our method is based on volumetric convolutions and molecular message passing, and as a result, the same network can be applied on different molecules. To demonstrate this, we have included an experiment where we train TDDFTNet on a mixed dataset containing all three molecules. The results are shown in Appendix F, which shows that TDDFTNet can be effectively trained on datasets containing multiple molecules.
>
>
> As we have discussed in our newly-added Open Directions appendix, the ability to generalize over molecules is a valuable task and will likely require a large-scale multi-molecule, multi-geometry TDDFT dataset, for which the computation required for generation will be quite high, as can be seen in our Time Comparison appendix. However, to get an initial idea of generalization ability, we evaluate a TDDFTNet model trained on the Ethanol molecule dataset on the Malondialdehyde molecule dataset. The results show that TDDFTNet can still produce correlated predictions within the first 5 steps, potentially thanks to the similarity in chemical composition between these molecules. However, the relative errors become close to 100% after 5 steps.
>
>
> | Time step                         | 1     | 2     | 3     | 4      | 5      | > 5           |
> | ----------------------------------|-------|-------|-------|--------|--------|---------------|
> | Dipole error ($1\times10^{-2}$)   | 34.12 | 56.73 | 79.47 | 94.16  | 79.17  | $\approx$ 100 |
> | Scaled-L2 error ($1\times10^{-2}$)| 77.84 | 83.14 | 98.42 | 104.39 | 105.23 | $\approx$ 100 |
>
>
> Additionally, we would like to note that defining a new learning task usually requires some extra effort. For example, learning would not be possible without properly scaling the small density difference. As a result, it is not uncommon for ML methods to be first developed for single molecules, and then be extended to more diverse molecules. As an example, the Hamiltonian prediction task was first developed for single molecules (Schütt et al., 2019, Unke et al., 2021, Yu et al., 2023) (e.g., the same MD17 data used in our work) and then extended to large-scale datasets with diverse molecules (Yu et al., 2024, Khrabrov et al., 2024). We aim to follow a similar path for learning TDDFT.
>
>
> Schütt, Kristof T., et al. "Unifying machine learning and quantum chemistry with a deep neural network for molecular wavefunctions." Nature communications 10.1 (2019): 5024.
>
> Unke, Oliver, et al. "SE (3)-equivariant prediction of molecular wavefunctions and electronic densities." Advances in Neural Information Processing Systems 34 (2021): 14434-14447.
>
> Yu, Haiyang, et al. "Efficient and equivariant graph networks for predicting quantum Hamiltonian." International Conference on Machine Learning. PMLR, 2023.
>
> Yu, Haiyang, et al. "Qh9: A quantum hamiltonian prediction benchmark for qm9 molecules." Advances in Neural Information Processing Systems 36 (2024).
>
> Khrabrov, Kuzma, et al. "$\nabla^ 2$ DFT: A Universal Quantum Chemistry Dataset of Drug-Like Molecules and a Benchmark for Neural Network Potentials." arXiv preprint arXiv:2406.14347 (2024).
>
> ## Q3
>
> > Q3. Please clarify what the three subplots in figure 4 is about.
>
> Figure 4 shows the predicted dipoles in the $x$ direction for each of the molecules, which is the direction with the most significant dipole responses. In the original version of our manuscript, the plots were mistakenly missing labels denoting the molecule name. We have added these labels in our revised manuscript -- thank you for bringing this to our attention. We have additionally added a more informative caption to this figure.

---

> ### Comment · Reviewer_t8k9 · 2024-11-26
>
> I appreciate the authors' efforts into answering my questions and the additional experiments. I do see the value of this work in terms of analyzing how to simulate TDDFT via neural network from the physical science perspective. But also as other reviewers are mentioned, the ML method itself is not novel and the training and evaluation were done with the same molecules and the number is limited. I see the additional experiments but I think they are not convincing enough to make up this weakness, and also I agree this may need time and is hard to be finished within the rebuttal period. As a overall considering, I raised my score to 5.

---

> ### Author Response · Authors · 2024-12-03
>
> We greatly appreciate your willingness to re-evaluate our work, as well as for your careful and insightful suggestions and feedback which have served to greatly improve our manuscript.
>
> > I see the additional experiments but I think they are not convincing enough to make up this weakness.
>
> The results presented in our previous reply mistakenly included metrics for a version of TDDFTNet trained for a shorter amount of time. These same results were also mistakenly included in Table 1 of the current version of our manuscript, however, the corrected results are in Table 2. After making this correction, we can see that the dipole loss for **TDDFTNet** surpasses that of **SphereNet-wd-coord**. The strong performance of SphereNet compared to SchNet has furthermore motivated us to add experiments replacing the distance-based message passing used by SchNet with SphereNet-based message passing.
>
> Specifically, both the encoder and the propagator use a 3-layer SphereNet with 256 hidden channels, so that the message passing part has similar network layers with SphereNet-wd-coord. We also tested the model with and without conditioning on atom coordinates in message passing, and the resulting models are dubbed **TDDFTNet-SphereNet-coord** and **TDDFTNet-SphereNet-w/o-coord**. Further, we tested the effect of removing the volume to atom message passing from **TDDFTNet-SphereNet-coord**, and the resulting model is dubbed **TDDFTNet-SphereNet-coord-w/o-vol2atom**. Finally, as shown in the ablation suggested by reviewer **EtBy**, using the dipole loss is helpful for making accurate dipole predictions. To futher push the accuracy of dipole prediction, we trained a model with a higher dipole loss weight (increased from 0.1 to 0.3), and the model is dubbed **TDDFTNet-SphereNet-coord-larger-dipole-loss**.
> The results are summarized in the below table, which we will integrate into the revised paper:
>
>
> | Model                                        | Dipole Error ($1\times10^{-2}$)$\downarrow$ | Scaled-$L_2$ Error ($1\times10^{-2}$) $\downarrow$ |Spectrum Overlap (\%)$\uparrow$ |
> |----------------------------------------------|---------------|------------|------------|
> | TDDFTNet                                     |   56.79       |  71.45     |  99.34     |
> | SphereNet-wd-coord                           |   58.26       |  N/A       |  99.39     |
> | TDDFTNet-SphereNet-coord                     |   55.98       |  72.00     |  99.41     |
> | **TDDFTNet-SphereNet-w/o-coord**          |   **54.21**  |  **70.54** | **99.46** |
> | TDDFTNet-SphereNet-coord-w/o-vol2atom        |   55.68       |  71.22     |  99.43     |
> | TDDFTNet-SphereNet-coord-larger-dipole-loss  |   53.94       |  74.87     |  99.48     |
>
> As can be seen, the performance improves notably when we use SphereNet-based message passing, giving better performance than **SphereNet-wd-coord**. These results show that the TDDFTNet architecture can effectively integrate information from the atomic structure to the volumetric feature map, which can result in improved property prediction compared to GNN-only models. We furthermore emphasize that explicit modeling of the density enables us to compute additional relevant properties, such as higher-order dipole moments, and thus, neural TDDFT not only enables performance gains relative to the direct prediction approach taken by the GNN models, but also enables computation of a wider range of properties without retraining.
>
> Second, **TDDFTNet-SphereNet-w/o-coord** shows better results than **TDDFTNet-SphereNet-coord**, which is rather surprising considering the previous GNN results. This can be explained by the fact that the initial density already contains the orientation information so coordinate information is not necessary to break the rotation-invariance.
>
> Third, **TDDFTNet-SphereNet-coord-w/o-vol2atom** shows better performance than TDDFTNet-SphereNet-coord. This suggests that volume to atom message passing may not be helpful in improving the performance for this task. We think that the substantial improvements from message passing shown in Table 2 of our manuscript is from the directed interaction from the atom graph to the volume feature map, which provides geometric information to the volume model. This highlights the effectiveness of one of our primary novelties, that is, the effective integration of convolution and message passing mechanisms.
>
> Finally, **TDDFTNet-SphereNet-coord-larger-dipole-loss** shows even better dipole prediction performance, at the cost of a worse density prediction accuracy. This is in line with our previous observation regarding the trade-off of using dipole loss, and it further shows the potential of TDDFTNet for property prediction compared to direct-prediction approaches.

---

> ### Author Response · Authors · 2024-12-03
>
> Last but not least, we would like to re-iterate the significance of developing methods for generalizing over multiple geometries of a single molecule. First, although the training and test set have the same type of molecule, the geometries in the test set are unseen during training. Such a scenario has wide applications such as in molecular dynamics.
>
> Second, as shown in the references cited in the response to W2 and Q1, developing machine learning methods first for single molecule, and then extending it to different molecules is a standard approach in the machine learning community. The demonstration of learning TDDFT on single molecule is solid step towards more general machine learning modeling of TDDFT.

---

> ### Comment · Reviewer_t8k9 · 2024-12-03
>
> Thank the reviewers for clarifying these points. Yes please correct these results in the manuscripts. I would keep my raised score for the work.

---

### Official Review · Reviewer_EtBy · 2024-11-02

**Soundness:** 3
**Presentation:** 3
**Contribution:** 3
**Rating:** 6
**Confidence:** 3

**Summary:**

This paper proposes a method for learning the time-dependent electron density, which is particularly significant for applications in biological chromophores, quantum transport, and plasmonic catalysis. The neural TDDFT model predicts the time evolution of electron density using an encoder and a propagator module in an autoregressive manner.

**Strengths:**

- The paper introduces a physics-aware approach that enhances model performance in a convincing manner.
- By leveraging a valuable benchmark dataset from computationally intensive simulations and evalutation metrics, this work sets up new tasks and has the potential for significant impact on the community.

**Weaknesses:**

- The paper does not include comparisons with 3D neural PDE models, such as GNO [1] and FNO [2], which could serve as valuable baselines.

- Unlike recent models for (static) electron density that aim to predict continuous densities [3,4,5,6], neural TDDFT relies on a grid-based training algorithm, which restricts the output to a fixed resolution.

### Minors
- The paper should include references to recent work on charge density estimation from the machine learning community [5,6].
- The Rydberg unit should ideally be written as "Ry" for consistency with most scientific literature; "Ryd" is less common. This and other notations should be introduced more clearly and kindly, given the interdisciplinary audience.
- Visualization of the electron density over time would aid in better understanding.
- In Equation (14), an equal sign seems to change to assignment sign.
- A statement on reproducibility and ethics should be included.

[1] https://arxiv.org/abs/2003.03485

[2] https://arxiv.org/abs/2010.08895

[3] https://arxiv.org/abs/2311.10908

[4] https://arxiv.org/abs/2011.03346

[5] https://arxiv.org/abs/2402.04278

[6] https://arxiv.org/abs/2405.19276

**Questions:**

- Does the output of electron density ensure normalization?
- How is optical absorption calculated? Additionally, the reason why optical absorption is important should be highlighted to enhance the understanding of this work.
- Why does the model rely on a grid-based approach, unlike coefficient-based or GNN-based methods?
- What is the impact of model error? How much error is acceptable to replace the conventional method, and is the model’s output sufficient in quality?
- Why is dipole loss included in the model’s loss function? Wouldn’t predicting the ground-state density difference accurately be enough?

**Details Of Ethics Concerns:**

While this paper addresses density functional theory calculation, which is highly valuable in chemistry, physics, and materials science, the author does not mention any concerns about the potential misuse of this technology. For instance, there could be risks of malicious applications, such as creating chemical weapons. Although such scenarios are unlikely based on current understanding, I believe all researchers working in AI for scientific applications should remain vigilant about these possibilities, given that AI tools can be used without specialized knowledge

---

> ### Author Response · Authors · 2024-11-25
>
> ## W1
>
> > W1. The paper does not include comparisons with 3D neural PDE models, such as GNO [1] and FNO [2], which could serve as valuable baselines.
>
> As one of the most prominent neural surrogate models for dynamics, it is certainly of interest to study replacing the U-Net in TDDFT with an FNO, since, as discussed by Gupta and Brandstetter (2022), these two architectures employ complementary approaches for processing multiscale information -- sequential (U-Net) and parallel (FNO). To compare these models we ran an experiment by replacing U-Net with FNO in TDDFTNet. The results are summarized in the below table. We will also include these results in the appendix of the revised paper. As can be seen, TDDFT+FNO underperforms TDDFT. We suggest two potential explanations for these results.
>
> |     | Dipole Error ($1\times10^{-2}$) $\downarrow$ | Scaled-$L_2$ Error ($1\times10^{-2}$) $\downarrow$  | Spectrum Overlap (\%) $\uparrow$ |
> |-----|----------------------------------------------|-----------------------------------------------------|----------------------------------|
> | **TDDFTNet**     |          58.76                 |       72.55                                 |         99.34                            |
> | **TDDFTNet+FNO** |         69.95                 |      79.77                                  |   98.66                                 |
>
>
> First, as discussed in Section 4.3, the prediction target at time $t$, $\Delta\rho_t=\rho_t-\rho_0$, is several orders of magnitude smaller than the initial density $\rho_0$. This is because the response of the density to the electromagnetic field is largely governed by small scale fluctuations around the initial density. In the frequency domain, these fluctuations manifest primarily in high frequencies.
> However, to manage model size and FLOPs, FNO only works with the lowest frequency modes and truncates the remaining high frequency modes to 0. Thus, for tasks wherein evolution occurs primarily in high frequencies, such as the one we consider here, models that excel in processing local dynamics, as is the case with U-Nets due to convolution with $3\times3\times3$ kernels, may be a better choice.
>
> Second, Fourier convolutions impose periodic boundaries on the function being convolved. Consider the density $\rho:[0,1]^3\to\mathbb{R}^{+}$ on the unit cube -- this periodicity implies that for all $x_0,y_0,x_1,y_1\in[0,1]$, FNO models the density  such that $\rho(x_0,y_0,0)=\rho(x_1,y_1,1)$. The same statement can be made for the boundaries along the $x$ and $y$ direction. However, modeling points on opposite boundaries adjacent to one another is an inaccurate assumption for this problem. In practice, this is handled by padding the input function with zeros and subsequently removing the padding following the convolutions, which is automatically done to our data since the non-zero regions of the input volumetric data representing the density is centered in the middle of the cube. Nonetheless, it is likely that the periodicity either has no effect or a negative effect on model quality.
>
> With respect to GNO, it has been our observation that in cases where the input data are on a regularly spaced grid, models such as U-Nets and FNOs which leverage this structure tend to outperform models which are designed for increased flexibility, such as GNO. Even in problems on irregular grids, leveraging convolutions through the use of interpolation has been shown to substantially outperform GNO -- see, for example, Table 2 (titled **Shape-Net Car dataset (3.7k mesh points)**) in [Li et al. (2024)](https://openreview.net/forum?id=86dXbqT5Ua&noteId=MiMUNcGOTL) as well as Table 1 (titled **Benchmark on the elasticity problem**) in [Li et al. (2023)](https://arxiv.org/abs/2207.05209).
>
> Gupta, Jayesh K., and Johannes Brandstetter. "Towards Multi-spatiotemporal-scale Generalized PDE Modeling." Transactions on Machine Learning Research (2022).
>
> Li, Zongyi, et al. "Fourier neural operator with learned deformations for pdes on general geometries." Journal of Machine Learning Research 24.388 (2023): 1-26.
>
> Li, Zongyi, et al. "Geometry-informed neural operator for large-scale 3d pdes." Advances in Neural Information Processing Systems 36 (2024).

---

> ### Author Response · Authors · 2024-11-25
>
> ## W2 and M1
>
> > W2. Unlike recent models for (static) electron density that aim to predict continuous densities [3,4,5,6], neural TDDFT relies on a grid-based training algorithm, which restricts the output to a fixed resolution.
> > M1. The paper should include references to recent work on charge density estimation from the machine learning community [5,6].
>
> Modeling DFT with mesh-independent representations is a valuable and related line of work. In Section 5 of our revised manuscript, we have added a discussion of this property with references to Cheng & Peng (2024), Xiang et al. (2024), Jørgensen & Bhowmik (2020), and Kim & Ahn (2024). With respect to TDDFTNet, this property can be can be inherited through the use of neural operators in place of U-Net components (Hao et al., 2023; Raonic et al., 2024). We discuss this further in the newly-added Open Directions section of our revised manuscript.
>
> Cheng, Chaoran, and Jian Peng. "Equivariant neural operator learning with graphon convolution." Advances in Neural Information Processing Systems 36 (2024).
>
> Fu, Xiang, et al. "A Recipe for Charge Density Prediction." arXiv preprint arXiv:2405.19276 (2024).
>
> Hao, Zhongkai, et al. "Gnot: A general neural operator transformer for operator learning." International Conference on Machine Learning. PMLR, 2023.
>
> Jørgensen, Peter Bjørn, and Arghya Bhowmik. "DeepDFT: Neural Message Passing Network for Accurate Charge Density Prediction." Machine Learning for Molecules Workshop@ NeurIPS 2020. 2020.
>
> Kim, Seongsu, and Sungsoo Ahn. "Gaussian Plane-Wave Neural Operator for Electron Density Estimation." Forty-first International Conference on Machine Learning (2024).
>
> Raonic, Bogdan, et al. "Convolutional neural operators for robust and accurate learning of PDEs." Advances in Neural Information Processing Systems 36 (2024).
>
> ## M2
>
> > M2. The Rydberg unit should ideally be written as "Ry" for consistency with most scientific literature; "Ryd" is less common. This and other notations should be introduced more clearly and kindly, given the interdisciplinary audience.
>
> In our revised manuscript, we have clarified the Rydberg unit, as well as Angstroms and electron-Volts. Thank you for pointing this out.
>
>
> ## M3
>
> > M3. Visualization of the electron density over time would aid in better understanding.
>
> In Appendix I, we have added a visualization of the electron density for malondialdehyde molecule, including the input density, the ground-truth density rollout, the predicted density rollout and the rollout error.
> The visualizations are produced in a way where we first crop the volume to the center 12 voxels along $y-$axis, and then sum the cropped volume along $y-$axis to produce a 2D plot.
> We will also include other molecules in the final version.
>
> ## M4
>
> > M4. In Equation (14), an equal sign seems to change to assignment sign.
>
> Thank you for pointing this out -- we have corrected this in our revised manuscript.
>
>
> ## M5
>
> > M5. A statement on reproducibility and ethics should be included.
>
> We have added a statement on reproducibility and ethics to Appendix B.
>
>
> ## Q1
>
>
> > Q1. Does the output of electron density ensure normalization?
>
> Yes. The initial electron density sums to the number electrons in a molecule (8 for water, 28 for malondialdehyde and 20 for ethanol).
> Due to the enforcement of density conservation in Equation 14, the total predicted density difference sums to zero at each time step and thus will not change the total initial density $\rho_0$. Since $\rho_0$ is normalized, the predicted densities must be as well due to this constraint. In our revised manuscript, we have stated the implications of this constraint more clearly at the end of Section 4.3.
>
> ## Q2
>
> > Q2. How is optical absorption calculated? Additionally, the reason why optical absorption is important should be highlighted to enhance the understanding of this work.
>
> The optical absorption is essentially defined as the ratio between the Fourier transform of the dipole and the Fourier transform of the electric field. The peaks in the absorption spectrum contains information about excited states of the molecule where the location of peak gives the excited state energy and the height of peak gives the oscillation strength. In practice, once we computed the time evolution of dipole, we can obtain the absorption spectrum using the [utility tool provided from the octopus software](https://www.octopus-code.org/documentation/15/manual/utilities/oct-propagation_spectrum/).
>
> We agree that a more detailed description of optical absorption is important to enhance the understanding. We will add more details with dedicated sections in the revised paper. Thank you for the suggestion.

---

> ### Author Response · Authors · 2024-11-25
>
> ## Q3.
>
> > Q3. Why does the model rely on a grid-based approach, unlike coefficient-based or GNN-based methods?
>
>
> It is indeed possible to describe TDDFT using local basis instead of real-space grid, in which case we can learn the coefficients for basis functions such as spherical harmonics. The benefit of using local basis representation is that it can greatly reduce the storage space, as only a set of coefficients need to be stored, as opposed to the volumetric data in real space.
> On the other hand, grid-based representation is more native to the time propagation of electronic wavefunction / density. In particular, the software we used to generate the dataset is primarily grid-based.
> We think coefficient-based or GNN-based method is an interesting direction to be explored in future work.
>
>
> ## Q4.
>
> > Q4. What is the impact of model error? How much error is acceptable to replace the conventional method, and is the model’s output sufficient in quality?
>
> This highly depends on the application and the properties which one desires to derive from the predicted density. In many applications, scientists are interested in the optical absorption spectrum, as it determines the excited energies of the geometry, which are an important factor in determining the viability of the geometry. As we have demonstrated, our model can accurately recover the optical absorption spectrum, and thus, we suggest that it has potential to be valuable to practitioners.
>
>
> ## Q5.
>
> > Q5. Why is dipole loss included in the model’s loss function? Wouldn’t predicting the ground-state density difference accurately be enough?
>
>
> This is certainly an interesting ablation -- thank you for the suggestion. We have added an experiment where we remove the dipole loss from the loss function. The results are summarized below. We can see that when the dipole loss is removed, the density prediction can be more accurate (as shown by the scaled-L2 loss). However, the dipole prediction becomes worse. This shows although predicting density accurately can in principle ensure predicting the correct dipole, adding additional loss is still beneficial to improve the prediction of global properties such as dipole.
>
>
> |     | Dipole Error ($1\times10^{-2}$) $\downarrow$ | Scaled-$L_2$ Error ($1\times10^{-2}$) $\downarrow$  | Spectrum Overlap (\%) $\uparrow$ |
> |-----|----------------------------------------------|-----------------------------------------------------|----------------------------------|
> | **TDDFTNet**                   |          58.76   |       72.55                                         |         99.34                    |
> | **TDDFTNet w/o dipole loss**  |         69.81   |      69.12                                           |      98.79                      |

---

> ### Comment · Reviewer_EtBy · 2024-11-26
>
> Thank you for your efforts and for providing additional clarifications. While the core methodology for addressing the problem may not be highly novel, the concepts surrounding the construction of electron densities in latent space and the physics-aware parameterization are compelling and merit presentation to the community. I will stand by my score.
>
> Additionally, I understand that we are nearing the end of the review process. However, I would like to suggest that incorporating an error analysis along the time domain could be highly beneficial for future researchers who wish to build upon this work.

---

> > ### Author Response · Authors · 2024-12-03
> >
> > Thank you for your careful feedback, which has served to improve our manuscript. We appreciate your recognition of our contributions as compelling and as having merit.
> >
> > Our newly added experiments in our latest reply to reviewer t8k9 demonstrate the effectiveness of our hybrid convolutional-message passing framework, which, in addition to the physics aware components, is one of the primary novelties of TDDFNet. Specifically, in these new experiments, we compare to direct prediction of the dipole moments using a GNN applied to the molecule geometry and find our TDDFTNet to improve performance.
> >
> > We have reported the error analysis along time domain in Figure 5. However, improving the stability of neural TDDFT models over long time-integration periods also remains a challenging and relevant open direction, which we will discuss in our Open Directions section in the Appendix.

---

### Official Review · Reviewer_XawS · 2024-11-04

**Soundness:** 3
**Presentation:** 3
**Contribution:** 3
**Rating:** 8
**Confidence:** 3

**Summary:**

This paper presented a novel machine learning (ML) approach for real-time time-dependent density functional theory (RT-TDDFT) simulations. Specifically, the authors modeled electronic density as volumetric data and incorporated a message-passing neural network (MPNN) for modeling atomistic interactions with ML. By training a deep neural network termed TDDFTNet with a part of the MD17 dataset, this paper showed that the proposed TDDFTNet can accurately and efficiently predict the time propagation of electron densities subject to an external pulse electromagnetic field.

**Strengths:**

### Originality
*I am not quite familiar with the recent literature on RT-TDDFT simulations. My reference comes from the `Related Work` section of this paper.*
1. The authors proposed an ML-based approach for RT-TDDFT simulations by discretizing the electronic density as volumetric data to simplify the representation of electron density as opposed to using basis sets in popular quantum chemistry theories.
1. The authors integrated message-passing for training TDDFTNet to model atomistic interactions.
1. The current literature on RT-TDDFT simulations is limited and this work provides a novel approach to address the challenges in RT-TDDFT simulations.

### Quality
1. The authors provided a detailed description of the problem setting and method in sections 3 and 4. Figure 2 is particularly helpful in representing the architecture of TDDFTNet.
1. The experiments and shown results are detailed and well-organized. The authors compared the performance of TDDFTNet on three molecules and compared the results with the ground truth from first-principle calculations by Octopus, an RT-TDDFT simulation software.

### Clarity
1. The paper is detailed in introducing the problem setting, method, the chosen experiment, and presenting the results.

### Significance
1. RT-TDDFT simulations are computationally expensive with first-principle calculations. Yet they are crucial for simulating molecules and modeling various physical properties. An efficient ML-based approach like TDDFTNet is significant for screening molecules and accelerating discoveries.
1. The TDDFTNet model is a good starting point for further research in RT-TDDFT simulations with ML.

**Weaknesses:**

**The training and evaluation were done with the same molecules.**
- The authors chose three molecules, water, malondialdehyde, and ethanol, from the MD17 dataset. For each molecule, 1600 geometries were sampled for training.
- However, the results of the optical absorption spectra and predicted dipoles were reported on the same three molecules. Even though the reported results could be from a different set of geometries, this still brings concerns about information leakage and generalization of the model to unseen molecules.
- To address this, the authors could introduce two other sets of experiments:
    1. Use different sizes for the training set geometry samples and evaluate the model on the same three molecules. For example, report the performance when training with 200, 400, 800, and 1600 geometries per molecule. If similar accuracy can be achieved with fewer samples, then TDDFTNet could still be useful even if it suffers generalization issues.
    1. Train the model on one set of molecules and evaluate it on another set of molecules.

**The discretization of the electronic density as volumetric data limits the scalability of the model.**
- The results shown are for three small molecules and the authors did not discuss the scalability of the model to larger molecules.
- For larger molecules, the discretization of the electronic density as volumetric data could be computationally expensive and may not be feasible since large molecules come with larger volumes.
- In addition, the volumetric representation brings concern with the grid size/fidelity of the representation. The authors should discuss the impact of the grid size on the performance of TDDFTNet.

**Questions:**

The questions are highly related to the weaknesses mentioned above. The authors should consider addressing the following questions in addition to the concerns shared in the weaknesses section:
1. In the experiment, the authors trained and evaluated TDDFTNet on the same molecule(s). The current implementation seems to suggest a combination of fewer first-principle calculations and TDDFTNet for RT-TDDFT simulations on the same molecule(s). How does TDDFTNet generalize to unseen molecules? Could the authors provide results on the generalization of TDDFTNet to unseen molecules?
1. Is the TDDFTNet model scalable to larger molecules such as toluene, caffeine, or even larger molecules? How does the volumetric representation of the electronic density impact the scalability of TDDFTNet to larger molecules?
1. Does the grid size/fidelity of the volumetric representation impact the performance of TDDFTNet? Did the authors experiment with different grid sizes and evaluate the performance of TDDFTNet?

---

> ### Author Response · Authors · 2024-11-25
>
> ## W1 and Q1
>
> > W1. The authors chose three molecules, water, malondialdehyde, and ethanol, from the MD17 dataset. For each molecule, 1600 geometries were sampled for training. However, the results of the optical absorption spectra and predicted dipoles were reported on the same three molecules. Even though the reported results could be from a different set of geometries, this still brings concerns about information leakage and generalization of the model to unseen molecules. To address this, the authors could introduce two other sets of experiments: Use different sizes for the training set geometry samples and evaluate the model on the same three molecules. For example, report the performance when training with 200, 400, 800, and 1600 geometries per molecule. If similar accuracy can be achieved with fewer samples, then TDDFTNet could still be useful even if it suffers generalization issues. Train the model on one set of molecules and evaluate it on another set of molecules.
>
> > Q1. In the experiment, the authors trained and evaluated TDDFTNet on the same molecule(s). The current implementation seems to suggest a combination of fewer first-principle calculations and TDDFTNet for RT-TDDFT simulations on the same molecule(s). How does TDDFTNet generalize to unseen molecules? Could the authors provide results on the generalization of TDDFTNet to unseen molecules?
>
>
> An ablation thoroughly probing the scaling laws of our model with respect to data volume would be an insightful addition. We have therefore added a study of this and the results are summarized in the below table. As can be seen, the performance only degrades slightly when the number of training samples is reduced from 1600 to 800, showing the good generalization ability of TDDFTNet with fewer samples. However, the performance degrades more when the number of training examples is further halved, which is potentially due to the complexity of the TDDFT propagation, even among the conformations of the same molecules. We will include this study in the appendix. Thank you for the suggestion.
>
>
> |  | Dipole Error ($1 \times 10^{-2}$) $\downarrow$ | Scaled-$L_2$ Error ($1 \times 10^{-2}$) $\downarrow$  | Spectrum Overlap (\%) $\uparrow$ |
> | -------- | -------- | -------- | --- |
> | TDDFTNet - 200 training samples   |  74.43    | 80.28    |  98.88   |
> | TDDFTNet - 400 training samples   |  66.96    | 75.14    |  99.12   |
> | TDDFTNet - 800 training samples   |  59.65    | 72.41    |  99.32   |
> | TDDFTNet - 1600 training samples  |  58.76    | 72.55    |  99.34   |
>
>
>
> Although our model is able to be trained on datasets with different molecules, as shown by the mixed dataset experiment in Appendix F, the ability to generalize to unseen molecules effectively will likely require the generation of a large scale TDDFT dataset spanning a variety of molecules and geometries, which we discuss further in the newly added Appendix on Open Directions. However, to get an initial idea of generalization ability, we evaluate a TDDFTNet model trained on the Ethanol molecule dataset on the Malondialdehyde molecule dataset. The results show that TDDFTNet can still produce correlated predictions within the first 5 steps, potentially thanks to the similarity in chemical composition between these two molecules. However, the relative errors become close to 100% after 5 steps.
>
>
> | Time step                         | 1     | 2     | 3     | 4      | 5      | > 5           |
> | ----------------------------------|-------|-------|-------|--------|--------|---------------|
> | Dipole error ($1\times10^{-2}$)   | 34.12 | 56.73 | 79.47 | 94.16  | 79.17  | $\approx$ 100 |
> | Scaled-L2 error ($1\times10^{-2}$)| 77.84 | 83.14 | 98.42 | 104.39 | 105.23 | $\approx$ 100 |

---

> > ### Author Response · Authors · 2024-11-25
> >
> > Additionally, we would like to note that due to the extra workload required when studying a new task, it is not uncommon for ML methods to be first developed for single molecules, and then be extended to more diverse molecules. As an example, the Hamiltonian prediction task was first developed for single molecules (Schütt et al., 2019, Unke et al., 2021, Yu et al., 2023) (e.g., the same MD17 data used in our work) and then extended to large-scale datasets with diverse molecules (Yu et al., 2024, Khrabrov et al., 2024). We aim to follow a similar path for learning TDDFT.
> >
> > Schütt, Kristof T., et al. "Unifying machine learning and quantum chemistry with a deep neural network for molecular wavefunctions." Nature communications 10.1 (2019): 5024.
> >
> > Unke, Oliver, et al. "SE (3)-equivariant prediction of molecular wavefunctions and electronic densities." Advances in Neural Information Processing Systems 34 (2021): 14434-14447.
> >
> > Yu, Haiyang, et al. "Efficient and equivariant graph networks for predicting quantum Hamiltonian." International Conference on Machine Learning. PMLR, 2023.
> >
> > Yu, Haiyang, et al. "Qh9: A quantum hamiltonian prediction benchmark for qm9 molecules." Advances in Neural Information Processing Systems 36 (2024).
> >
> > Khrabrov, Kuzma, et al. "$\nabla^ 2$ DFT: A Universal Quantum Chemistry Dataset of Drug-Like Molecules and a Benchmark for Neural Network Potentials." arXiv preprint arXiv:2406.14347 (2024).

---

> ### Author Response · Authors · 2024-11-25
>
> ## W2 and Q2
>
> > W2. The discretization of the electronic density as volumetric data limits the scalability of the model. The results shown are for three small molecules and the authors did not discuss the scalability of the model to larger molecules. For larger molecules, the discretization of the electronic density as volumetric data could be computationally expensive and may not be feasible since large molecules come with larger volumes. In addition, the volumetric representation brings concern with the grid size/fidelity of the representation. The authors should discuss the impact of the grid size on the performance of TDDFTNet.
>
> > Q2. Is the TDDFTNet model scalable to larger molecules such as toluene, caffeine, or even larger molecules? How does the volumetric representation of the electronic density impact the scalability of TDDFTNet to larger molecules?
>
> Improved scalability does represent a valuable open-direction which will be relevant when working with more complex systems, and thus, we have added a discussion of this point to the newly-added Open Directions section in the appendix of our revised manuscript. On the other hand, TDDFTNet training was conducted on 4 2080RTX GPUs (each with 12GB VRAM), and therefore, we expect that the current approach can be scaled to systems moderately larger than those which we have considered here. Regarding the relation of performance and grid spacing, please see our reply to Q3 below.
>
> To have an concrete idea about the scalability, we ran Octopus for a caffeine molecule using the 3D structure from [PubChem](https://pubchem.ncbi.nlm.nih.gov/compound/Caffeine). The grid produced by the TDDFT simulation is $85\times75\times49$, which is larger than the grid in our dataset, which is about $50\times50\times50$. With the increased grid, we would expect the computation cost to be higher. However, since the size of the grid increases only for certain dimensions, we expect the computation cost to remain manageable, perhaps with a doubled GPU memory and/or computational time. In fact, the way octopus creates the grid is to first create a sphere around each atom, with a radius equals to 3.5 angstrom in our case, and then create a box which covers all spheres. As a result, the size of the simulation box depends on the shape of the molecule, and the size of the grid will not increase significantly unless the molecule has large spatial extents in all 3 spatial direction, which we believe is uncommon for organic molecules.
>
>
> Nevertheless, beyond grid-based representation, it is also possible to describe TDDFT using local basis, where the wavefunction / density can be represented as coefficients for basis functions such as spherical harmonics. The benefit of using local basis representation is that it can greatly reduce the storage space, as only a set of coefficients need to be stored, as opposed to the volumetric data in real space. On the other hand, grid-based representation is more native to the time propagation of electronic wavefunction / density. In particular, the software we used to generate the dataset is primarily grid-based. We think using the local basis representation is an interesting direction to be explored in future work.
>
>
>
> ## Q3
>
> > Q3. Does the grid size/fidelity of the volumetric representation impact the performance of TDDFTNet? Did the authors experiment with different grid sizes and evaluate the performance of TDDFTNet?
>
> With respect to the spatial grid spacing, we use [the grid spacing recommended by Octopus](https://www.octopus-code.org/documentation//15/tutorial/response/optical_spectra_from_time-propagation/?series=optical-response). The spatial grid size for training TDDFTNet is kept the same as for data generation, and we expect that coarsening would reduce performance, although it certainly would increase speed.
>
> The time spacing is $50\times$ that which used to generate the data, and we again expect a similar speed-accuracy tradeoff.

---

> > ### Comment · Reviewer_XawS · 2024-11-26
> >
> > Thank the authors for the discussions.
> >
> > A high-quality MD dataset is indeed difficult to get. For future directions, I believe that looking into ways of increasing the scalability of larger molecules and transferability to different molecules can greatly further the significance of the work. The idea of using grid points is simple and elegant and potential could lead to many other advancements in the future.
> >
> > I will thus keep my score.

---

### Official Review · Reviewer_6yvB · 2024-11-04

**Soundness:** 3
**Presentation:** 4
**Contribution:** 3
**Rating:** 8
**Confidence:** 3

**Summary:**

The authors propose a new approach to accelerating RT-TDDFT simulations using machine learning. The authors design a novel neural network architecture called TDDFTNet that incorporates the 3D geometry of the molecule and uses a latent evolution approach to model the changes in electron density over time. The novelty of this method stems from the message passing architecture and time evolution in a physics-aware latent space. The method is evaluated on datasets of molecules from the MD17 dataset and shows promising results in accurately predicting the evolution of electron density and the optical absorption spectrum.

**Strengths:**

- The proposed TDDFTNet architecture is well-motivated and incorporates key physical constraints to ensure the model learns physically meaningful representations.
- The use of a latent evolution approach is novel in this context and allows the model to capture long-term dependencies in the data.
- The experimental results demonstrate the effectiveness of the proposed method in accurately predicting the evolution of electron density and the optical absorption spectrum.
- The ablation studies provided show a systematic improvement in results as different components of TDDFTNet are added.

**Weaknesses:**

- The paper could benefit from a more detailed discussion of the limitations of the proposed method and potential future directions. For example, how can the method be extended to handle larger systems or different types of excitations?
- The evaluation is limited to a small set of molecules from the MD17 dataset. It would be beneficial to see how the method performs on a larger and more diverse dataset, including more complex molecules and materials.
- The paper would benefit from a more detailed comparison with the numerical simulation computation time. How well does this method perform against Octopus? How is the speed up if you run Octopus with a similar coarser grid?

**Questions:**

- The paper primarily evaluates TDDFTNet on small organic molecules. How well do you expect this method to generalize to larger and more complex systems, such as macromolecules or periodic systems? What modifications or extensions might be necessary to handle such cases?
-  The current work focuses on simulating the response to an impulse electric field. Can TDDFTNet be extended to study other excited state dynamics, such as those induced by different types of perturbations or those involving non-adiabatic transitions? Can we incorporate perturbation parameters or a continuously time dependent external potential in the model? How well would it generalize
- As mentioned in weakness, a more detailed comparison with Octopus would improve the paper. How are the computational costs if you incorporate training costs? This paper highlights the common pitfalls in ML-PDE works and might be a good framework for comparing TDDFTNet and Octopus: https://www.nature.com/articles/s42256-024-00897-5

---

> ### Author Response · Authors · 2024-11-25
>
> ## W1
>
> > W1. The paper could benefit from a more detailed discussion of the limitations of the proposed method and potential future directions. For example, how can the method be extended to handle larger systems or different types of excitations?
>
> The introduction of neural TDDFT has opened the door for many new lines of work. In the newly-added section on Open Directions in the appendix of our revised manuscript, we have discussed several of these opportunities, including larger systems and different types of excitations.
>
>
>
> ## W2 and Q1
>
> > W2. The evaluation is limited to a small set of molecules from the MD17 dataset. It would be beneficial to see how the method performs on a larger and more diverse dataset, including more complex molecules and materials.
>
> > Q1. The paper primarily evaluates TDDFTNet on small organic molecules. How well do you expect this method to generalize to larger and more complex systems, such as macromolecules or periodic systems? What modifications or extensions might be necessary to handle such cases?
>
>
> In the newly added "Open Directions" section of the appendix, we have discussed how, while it will be computationally expensive to generate a machine-learning scale TDDFT dataset for such systems, it represents a promising direction where substantial gains can be realized with a trained model. In the case of periodic system, an extension could involve the use of architectures leveraging symmetries of periodic structures, such as crystals, to encode the molecule, and a U-Net model with periodic padding to encode the density.
>
>
>
> ## W3 and Q3
>
> > W3. The paper would benefit from a more detailed comparison with the numerical simulation computation time. How well does this method perform against Octopus? How is the speed up if you run Octopus with a similar coarser grid?
>
> > Q3. As mentioned in weakness, a more detailed comparison with Octopus would improve the paper. How are the computational costs if you incorporate training costs? This paper highlights the common pitfalls in ML-PDE works and might be a good framework for comparing TDDFTNet and Octopus: https://www.nature.com/articles/s42256-024-00897-5
>
> In Appendix D, we compare the computation time for Octopus and TDDFTNet and find that TDDFTNet offers great acceleration -- the time for solving TDDFT for a single Ethanol molecule is more than 15 minutes using Octopus versus less than 2 seconds using TDDFTNet. In order to comprehensively compute the overall time savings of neural TDDFT, we must know how many geometries the trained model will be used to screen. In many practical applications of TDDFT, the resulting density is highly sensitive to perturbations to the geometry, thereby requiring a large number of simulations with the same molecule in order to conduct a thorough study of the electronic structure. In such cases, the dramatic speed-up of TDDFTNet observed in Appendix D outweighs training time, which in our experiments typically took 2-3 days on 4 RTX2080 GPUs.
>
> With respect to the spatial and temporal grid spacing, we use [the grid spacing recommended by Octopus](https://www.octopus-code.org/documentation//15/tutorial/response/optical_spectra_from_time-propagation/?series=optical-response). The spatial grid size for training TDDFTNet is kept the same as for Octopus. We increase the time step size $50\times$ compared to that which used to generate the data. In general, taking too large of a time step size for data generation leads to a time evolution which diverges from the smaller step-size solution and can often lead to instability and blow-up of the solution. By contrast, we can almost always achieve a time savings using neural methods by temporally coarsening the solution produced by the numerical solver.
>
>
>
> ## Q2
>
> > Q2. The current work focuses on simulating the response to an impulse electric field. Can TDDFTNet be extended to study other excited state dynamics, such as those induced by different types of perturbations or those involving non-adiabatic transitions? Can we incorporate perturbation parameters or a continuously time dependent external potential in the model? How well would it generalize
>
> Certainly we could integrate various perturbation parameters using FiLM-style embeddings (Perez et al., 2018) in our U-Net, and similar feature modulations for the GNN embeddings. A similar approach could be done by using an embedding of a discretization of the external potential. We have discussed this further in the newly-added Open Directions section of the Appendix of our revised manuscript.
>
> Perez, Ethan, et al. "Film: Visual reasoning with a general conditioning layer." Proceedings of the AAAI conference on artificial intelligence. Vol. 32. No. 1. 2018.

---

> > ### Comment · Reviewer_6yvB · 2024-11-26
> >
> > Thank you for responding to my review. I am satisfied with the new sections in the appendix and believe that the current work opens up a rich line of inquiry for ML accelerated TDDFT calculations. I am keeping my score the same.

---

### Meta-Review · Area_Chair_TigM · 2024-12-20

**Metareview:**

The authors present a method for accelerating TD-DFT calculations for excited states of quantum systems by training a neural network on TD-DFT calculations from one set of geometries of a molecule and generalizing to other geometries of the same molecule. Their results suggest that the loss in accuracy relative to the ground truth is limited on the three molecules from MD17 they investigated. The proposed method is interesting, and tackles an important problem, but the experimental comparisons are still lacking. There is no baseline comparison of any sort except for some ablation studies, not only against related methods, but against trivial alternatives like KNN interpolation. The authors have not actually demonstrated that the problem of generalizing from one geometry to another on the same molecule is not well handled by simpler alternatives. Given this, I recommend against acceptance.

Another minor point: the authors describe the TDDFT training data as "ground truth", but ground truth would generally refer to experimental results. If experimental results are available, it would be good to show how well both their method and TDDFT do in comparison, as TDDFT itself is known to be not very accurate.

**Additional Comments On Reviewer Discussion:**

The reviewers all raised reasonable concerns about how well the method generalizes and the weakness of the baseline comparisons. Given this, I am surprised by how high a score some of them gave the paper.

---

### Decision · Program_Chairs · 2025-01-22

Reject